# Chamomile (*Matricaria chamomilla* L.): A Review of Ethnomedicinal Use, Phytochemistry and Pharmacological Uses

**DOI:** 10.3390/life12040479

**Published:** 2022-03-25

**Authors:** Amina El Mihyaoui, Joaquim C. G. Esteves da Silva, Saoulajan Charfi, María Emilia Candela Castillo, Ahmed Lamarti, Marino B. Arnao

**Affiliations:** 1Department of Plant Biology (Plant Physiology), Faculty of Biology, University of Murcia, 30100 Murcia, Spain; elmihyaoui.amina@gmail.com (A.E.M.); mcandela@um.es (M.E.C.C.); 2Laboratory of Plant Biotechnology, Department of Biology, Faculty of Sciences, Abdelmalek Essaadi University, Tetouan 93000, Morocco; lamarti.ahmed58@gmail.com; 3CIQ(UP)—Research Center in Chemistry, DGAOT, Faculty of Sciences, University of Porto, Rua do Campo Alegre, s/n, 4169-007 Porto, Portugal; jcsilva@fc.up.pt; 4Biology and Health Laboratory, Department of Biology, Faculty of Science, Abdelmalek Essaadi University, Tetouan 93000, Morocco; sawlajan@gmail.com

**Keywords:** chamomile, ethnomedicine, Matricaria, medicinal herbs, pharmacological activity, phytotherapy

## Abstract

*Matricaria chamomilla* L. is a famous medicinal plant distributed worldwide. It is widely used in traditional medicine to treat all kinds of diseases, including infections, neuropsychiatric, respiratory, gastrointestinal, and liver disorders. It is also used as a sedative, antispasmodic, antiseptic, and antiemetic. In this review, reports on *M. chamomilla* taxonomy, botanical and ecology description, ethnomedicinal uses, phytochemistry, biological and pharmacological properties, possible application in different industries, and encapsulation were critically gathered and summarized. Scientific search engines such as Web of Science, PubMed, Wiley Online, SpringerLink, ScienceDirect, Scopus, and Google Scholar were used to gather data on *M. chamomilla*. The phytochemistry composition of essential oils and extracts of *M. chamomilla* has been widely analyzed, showing that the plant contains over 120 constituents. Essential oils are generally composed of terpenoids, such as α-bisabolol and its oxides A and B, bisabolone oxide A, chamazulene, and β-farnesene, among other compounds. On the other hand, *M. chamomilla* extracts were dominated by phenolic compounds, including phenolic acids, flavonoids, and coumarins. In addition, *M. chamomilla* demonstrated several biological properties such as antioxidant, antibacterial, antifungal, anti-parasitic, insecticidal, anti-diabetic, anti-cancer, and anti-inflammatory effects. These activities allow the application of *M. chamomilla* in the medicinal and veterinary field, food preservation, phytosanitary control, and as a surfactant and anti-corrosive agent. Finally, the encapsulation of *M. chamomilla* essential oils or extracts allows the enhancement of its biological activities and improvement of its applications. According to the findings, the pharmacological activities of *M. chamomilla* confirm its traditional uses. Indeed, *M. chamomilla* essential oils and extracts showed interesting antioxidant, antibacterial, antifungal, anticancer, antidiabetic, antiparasitic, anti-inflammatory, anti-depressant, anti-pyretic, anti-allergic, and analgesic activities. Moreover, the most important application of *M. chamomilla* was in the medicinal field on animals and humans.

## 1. Introduction

*Matricaria chamomilla*, usually referred to as chamomile, is a well-known medicinal plant from the Asteraceae family. It is an annual herb that grows on all soil types and is resistant to cold. *M. chamomilla* is native to southern and eastern Europe and northern and western Asia [1]. Nowadays, it is widely distributed all around the world [2]. *M. chamomilla* has been used traditionally in several countries to cure a number of diseases, including gastrointestinal disorders [3], common cold [4], liver disorders [5], neuropsychiatric and respiratory problems [6]. Also, this plant is widely used against pain and infections [7] and to cure skin, eye, and mouth diseases [8] (Figure 1).

The phytochemical composition of *M. chamomilla* essential oil (EO) and extracts has been reported, with over 120 constituents identified. In general, terpenoids formed the most important compound group in *M. chamomilla* EO, with the most important compounds being bisabolol and its oxides A and B, bisabolone oxide A, chamazulene, and β-farnesene (Figure 2). This composition is influenced by several factors, such as the geographic regions and environment [9,10], plant cultivars [11], and genetic factors [12]. Other factors related to the drying techniques [13], extraction techniques [14], salicylic acid concentrations [11], and the use of cyanobacterial suspensions as bio-fertilizers [15] can also influence EO chemical composition. On the other hand, *M. chamomilla* extracts were dominated by phenolic compounds, including phenolic acids, flavonoids, and coumarins. In addition, the amino acid composition has also been studied [16]. Their chemical composition of coumarin, phenolic acid, and flavonoids contents was influenced by treatment with ethephon [17], cadmium, and copper [18].

Pharmacological investigations reported that *M. chamomilla* has several biological activities (Figure 1). The antioxidant activity of EO and extracts was investigated using several tests. Moreover, enzyme activities of extracts were assessed for catalase, acetylcholine esterase, glutathione, peroxidase, ascorbate peroxidase, and superoxide dismutase. In addition, the antioxidant activity determined in cell suspension culture [19], extracts from waste after chamomile processing [20], and polyphenolic–polysaccharide conjugates [21] was investigated. On the other hand, *M. chamomilla* exhibited an antibacterial potential against Gram-positive and Gram-negative bacteria. In addition, the effect of *M. chamomilla* EO on *Pseudomonas aeruginosa* biofilm formation and alginate production was investigated [22]. The plant also showed activity against antibiotic-resistant bacteria, including *Staphylococcus aureus* MRSA [23,24] and multidrug-resistant *P. aeruginosa* [25]. Moreover, *M. chamomilla* extract exhibited an anti-adherence activity against several bacterial strains [26]. On the other hand, *M. chamomilla* EO and extracts exhibited antifungal activity against several fungal strains, especially against *Candida* sp. and *Aspergillus* sp. Generally, *M. chamomilla’s* effect on oxidative stress, bacterial, and fungal strains varied depending on several factors, including plant origin, organ used, extraction solvent, and technique. In addition, *M. chamomilla* exhibited antiparasitic, insecticidal [27,28], anti-diabetic [29], anticancer [30,31], and anti-inflammatory activities [32] (Figure 1).

Based on the wide range of pharmacological activities demonstrated by *M. chamomilla,* its possible use in several fields has been investigated. The most important application of *M. chamomilla* was in the medicinal field. Indeed, several studies on animal models and patients showed the therapeutic effect of this plant on a wide range of diseases, including nervous diseases [33], reproductive diseases [34], diabetes [35], obesity and related metabolic disorders [36], cardiovascular diseases [37], gastrointestinal diseases [38,39,40], allergies [41], skin diseases [42], eye diseases [43], and mouth problems [44]. The plant also allowed pain-relieving [45], wound healing [46], and acted as a protective agent for kidney and liver [47], gastrointestinal [48], and reproductive systems [49]. On the other hand, *M. chamomilla* can be used as an anesthetic in aquaculture [50], supplementary animal feed [51], and food industry [52], as antifungal in agriculture [53], as a surfactant agent in chemical enhanced oil recovery [54], and as an anti-corrosive agent in federated mild steel [55]. In addition, *M. chamomilla* EOs and/or extracts have been encapsulated into silica nanoparticles [56], silver nanoparticles [57], chitosan nanocapsules [58], and alginate microcapsules [59]. This encapsulation allowed the enhancement of their properties such as anti-cancer [60], antiparasitic [58], catalytic [61], antibacterial, and antifungal activities [56,57,62], and also can improve their use as food additive [63].

## 2. Botanical and Ethnomedicinal Use

### 2.1. Taxonomy and Synonym

*Matricaria chamomilla* L. is a well-known medicinal plant from the Asteraceae family that has been called the “star among medicinal species” [64]. *M. chamomilla* (synonym *Matricaria recutita* L. Rauschert and *Chamomilla recutita*) is an old-time drug famously known as chamomile, German chamomile, Roman chamomile, Hungarian chamomile, and English chamomile [65,66]. The real chamomile is frequently mistaken with plants belonging to the Anthemis genus, especially *Anthemis cotula* L., a toxic plant with a terrible odor [67].

### 2.2. Botanical and Ecology Description

*M. chamomilla* is an annual herb with thin, spindle-shaped roots. The branched, erect stem grows to a height of 10 to 80 cm. The narrow and long leaves are bi- to tripinnate. *M. chamomilla* flower heads are pedunculate, heterogamous, separately placed with a diameter of 10 to 30 mm. The golden yellow tubular florets with 5 teeth are 1.5 to 2.5 mm long, always ending in a glandulous tube. The 11 to 27 white plant flowers are 6 to 11 mm long, 3.5 mm wide, and arranged concentrically. The receptacle is 6 to 8 mm wide, flat in the beginning, and conical. The fruit is a yellowish-brown cypsela with 3–5 faint ribs [1,2]. *M. chamomilla* can be grown on any type of soil, but growing the crop in rich, heavy, and damp soils should be avoided. It can grow at temperatures ranging from 7 °C to 26 °C and annual rainfall of 400 to 1400 mm per season. The plant can withstand cold weather but grows better in full sun and requires long summer days and high temperatures for optimum EO yield [68]. *M. chamomilla* is a diploid cell (2n = 18), allogamous in nature, starts blooming from the second week of March, and exhibits wide segregation as a commercial crop [2,69].

### 2.3. Geographic Distribution

*M. chamomilla* is native to southern and eastern Europe and northern and western Asia. It has also been introduced in many countries and has been naturalized in Britain, Australia, and North America [1,70]. Nowadays, it is widely distributed and cultivated. The plant is grown in Germany, Hungary, France, Yugoslavia, Russia, Brazil, New Zealand, and North Africa [2]. In Morocco, *M. chamomilla* is located in two separate areas, the first between Tangier, Ouezzane, Souk Larbaa, Moulay Bousselham, and Azilah, and the second between Kenitra, Sidi Slimane, Khemisset, and Rabat [71].

### 2.4. Ethnomedicinal Use

*M. chamomilla* is one of the most known plants for its traditional medicinal uses (Table 1). The traditional application of *M. chamomilla* depends on the plant parts (flower, leaves, stem, and whole plant) and the preparation methods (infusion, decoction, vapor inhalation, bath, and compress).

In traditional Moroccan medicine, several studies from different regions reported that flowers of *M. chamomilla* (Babonj/Babounj) represent the most used part, followed by leaves and whole plant. It is prepared as an infusion or decoction for the treatment of diabetes [72,73], nervous disorders, diarrhea, angina, canker sore, abscess, infections, and painful menstruation [7,74,75]. In Spain, *M. chamomilla* is used as an infusion against several diseases, including gastralgia, digestive disorder, female genital infection, and kidney stones [76]. In addition, the plant can be used as a sedative, antiseptic, antiemetic against nausea, and anti-inflammatory against gastric and intestinal disorders, also in eye irritations [77]. In Portugal, *M. chamomilla* flowering top is traditionally used against several diseases, including sciatic pain and infection of the mouth, throat, ear, and skin [6]. In Turkey, the infusion is used against colic spasm, cold, and as a sedative [4]. In Italian traditional medicine, *M. chamomilla* has been widely used against sprain, broken bones, irritability, and muscular or gastrointestinal pain [3]. In addition, it has been used as a sedative [78,79] and as a yellow dye, and for bleaching hair [79]. In Serbia, the infusion of *M. chamomilla* is used to strengthen the immune system and treat burns, stomach disorders, vaginal disorders, liver disorders, skin, and mucus inflammation. In addition, the infusion is used in skin, eye, and mouth care, and as an aroma for shampoos [5,80,81,82,83,84]. On the other hand, *M. chamomilla* is used in Greece to treat a number of gastrointestinal disorders, skin problems, and eye infections [8]. On the other hand, *M. chamomilla* tea is used in southeastern Albania to treat cough, diarrhea, and intestinal discomfort [85]. In Bulgaria, the plant is used against cold, throat pain, genitalia, swollen eyes, and for cleansing the face [86].

**Table 1 life-12-00479-t001:** Ethnomedicinal use of *M. chamomilla*.

Area of Study/Country	Local Name	Part Used	Mode of Preparation	Traditional Use	References
Taza region (Morocco)	Babonj	Flower	Infusion Decoction	Diabetes	[72]
Beni Mellal (Morocco)	Babounj	Flower	Infusion	Diabetes mellitus	[73]
Daraa-Tafilalet region, Errachidia province (Morocco)	Not specified	Whole plant	Infusion	Nervous disorders	[74]
Tarfaya Province (Morocco)	Babounj	Leaves	Decoction	Antispasmodic	[75]
Fez (Morocco)	Babounj	Flower	Infusion Decoction	Colic, diarrhea, nervousness, depression, angina, canker sore, painful menstruation, fever, abscess, infections	[7]
Hatay Province (Turkey)	Babuneç Papatya	Flower head	Infusion	Cold, colic spasm, sedative	[4]
Granada province (southern Spain)	Not specified	Flowery plant	Infusion	Gastralgia, digestive disorder, conjunctivitis, dysmenorrhea, cold, cough, gases, female genital infection, kidney stones, eye infection, headache, insomnia	[76]
Alt Empordà region (Catalonia, Spain)	Camamilla, Camamilla de jardí	Not specified	Decoction Infusion	Antiseptic, ocular antiseptic, conjunctivitis, digestive, gastric and intestinal anti-inflammatory, stomachache, nausea, antiemetic	[77]
Trás-os-Montes (northern Portugal)	Camomila	Flowering top	Infusion Decoction	Depression, nervousness, stress, insomnia, neuralgia, sciatic pain, digestive, stomachache, gases, intestinal colic, infection of mouth, throat, and ear, cellulitis, asthma	[6]
Island of Procida (Campania, southern Italy)	Cammumilla	Whole plant Stem Inflorescence	Infusion Decoction In the bath	Cold, cough, sprain, broken bones, irritability, tired eyes, conjunctivitis, abdominal colic, gastrointestinal pain, muscular pain, menstrual pain	[3]
National Park of Cilento and Vallo di Diano (southern Italy)	Hammamilla	Aerial parts	Infusion	Sedative, dye yellow, bleach hair	[79]
Monte Sicani Regional Park Central Western Sicily (southern Italy)	Kamilica	Flower	Infusion	Sedative, digestive	[78]
Bulgaria	Not specified	Flower head	Decoction Inhalation Compress	Throat pain, cold, swollen eyes, cleansing face, genitalia	[86]
Southeastern Albania	Kamilica Koromil	Flowering tops	Tea	Intestinal discomfort, diarrhea, cough	[85]
Pirot County (eastern Serbia)	Bela rada, kamilica, podrumce	Flower	Infusion	Stomach disorder	[80]
Rtanj Mt. (eastern Serbia)	Kamilica	Flower	Infusion	Immune system strengthening, cold, sedative, conjunctivitis, anti-inflammatory	[84]
Negotin Krajina (eastern Serbia)	Kamilica	Flower Leaves	Infusion	Digestive disorder, vaginal disorder, eye care skin care, aroma for shampoos	[81]
Jablanica district (southeastern Serbia)	Kamilica	Herb	Infusion	Skin and mucus inflammation, digestive disorder, cough, anxiety, eyewash, mouthwash	[82]
Zlatibor district (southwestern Serbia)	Not specified	Herb	Infusion	Cold and stomach disorder	[83]
Pčinja district (southeastern Serbia)	Kamilica	Herb	Infusion	Skin inflammation, burns, digestive disorder, liver disorder, cough, anxiety, insomnia, eyewash, mouthwash	[5]
Peloponnisos (Greece)	Hamomili	Inflorescences	Infusion Compress Vapor inhalation	Stomach disorder, constipation, ulcer, colic, allergy, insomnia, migraine, stress, skin problems (inflammation, dermatitis, acne, burn, eczema, itching, wound antiseptic), catarrh, sore throat, eye infection, aphthae, gingivitis, eyewash, mouthwash	[8]

## 3. Phytochemical Interest

The phytochemical composition of *M. chamomilla* EO and extracts has been widely investigated, and more than 120 constituents have been identified. Due to the large number of investigations found in this section, we have reported only the major studies reflecting the chemical composition in different environmental areas (Table 2). The phytochemical screening of EO and extracts was carried out using chromatographic techniques and focused on the flowers since they are the most used plant organ. Generally, the chemical composition of EO and extracts showed the abundance of terpenoids (Figure 2) and phenolic compounds, commonly phenolic acids (Figure 3), flavonoids (Figure 4), and coumarins (Figure 5).

Generally, the chemical composition varied significantly depending on the origin of the plants. The EO from Moroccan *M. chamomilla* obtained by microwave-assisted hydrodistillation showed 24 chemical components representing 98.49% of the total EO, with chamazulene (26.11%) as the main component, followed by *cis*-β-farnesene (11.64%) and eucalyptol (8.19%) [87]. In another study from Egypt, Abbas et al. [13] compared the chemical composition of EO obtained from fresh and dried flowers using different techniques (sunlight, shade, oven, solar dryer, and microwave). The findings of this work showed that the main component of all EO was α-bisabolol oxide A (33–50.5%). The drying methods used in this study significantly influenced the number of compounds identified, with 21 compounds found after solar drying and only 13 found after microwave drying. On the other hand, EL-Hefny et al. reported *cis*-β-farnesene (27%) as the main component of Egyptian EO, followed by D-limonene (15.25%) and α-bisabolol oxide A (14.9%) [88]. In another study on *M. chamomilla* from Turkey, the EOs obtained from plants cultivated in two locations (Konya and Karaman) showed a quantitative difference in their chemical composition, with a dominance of α-bisabolol (27.36–38.6%), β-farnesene (25.05–30.15%), and chamazulene (13.5–13.93%) [89]. Furthermore, Berechet et al. found that *M. chamomilla* EO from Romania is composed mainly of sesquiterpenes (91.65%), dominated by bisabolol oxide A (70.2%) [90]. Brazilian *M. chamomilla* EO was formed of 18 compounds, mainly α-bisabolol oxide B (26.08%), β-farnesene (16.35%), and bisabolol oxide A (14.7%) [91]. The EOs extracted from 13 cultivated Italian *M. chamomilla* were mainly composed of *cis*-tonghaosu (11.8–45.9%) or α-bisabolol oxide B (3.7–28.1%) [92]. In general, these results showed high variation in component quantity in EOs, which is attributed to the effect of the environment on plant development. On the other hand, several studies reported the chemical composition of Iranian *M. chamomilla* EO. Generally, the EO was dominated either by α-bisabolol oxide A [11,12,14,15,93,94,95,96,97,98], α-bisabolone oxide A [10,12], bisabolol oxide A [11], α-bisabolol [96], bisabolol oxide B [11], α-bisabolol oxide B [12], α-bisabolol oxide B and chamazulene [10], chamazulene [95], *cis*-pinocamphone [14], *trans*- and *cis*-γ-bisabolene [10], or trans-β-farnesene [94]. This composition varied depending on the extraction technique [14], geographical factors [10], environmental conditions (normal or heat stress), plant cultivar, and salicylic acid concentrations and their interaction [11], and the use of cyanobacterial suspensions as bio-fertilizer [15]. On the other hand, the salinity of irrigation water did not significantly affect the EO quantity and composition, and also apigenin content [93]. In a study, Mavandi et al. carried out a comparison between EOs extracted from six native accessions of Iranian *M. chamomilla* and two German and Hungarian varieties. The results revealed that the Iranian EOs belonged to three chemotypes: α-bisabolol oxide A, α-bisabolol oxide B, and α-bisabolone oxide A. The EOs from Germany and Hungary belonged to α-bisabolol oxide A chemotype. This study showed the effect of genetic factors on EO composition [12]. In a study on *M. chamomilla* from 11 countries, Orav et al. reported different compositions depending on geographic regions and environmental factors, with EOs from Germany, Estonia, Greece, Scotland, England, and Latvia dominated by bisabolol oxide A chemotype, EOs from Moldova, Russia, and Czech dominated by α-bisabolol chemotype, EO from Armenia dominated by bisabolol oxide B chemotype, and EO from Ukraine dominated by bisabolone oxide A chemotype [9]. On the other hand, *M. chamomilla* EO from Australia is composed mainly of bisabolol oxides B and A (27.5 and 27%, respectively) [99]. Furthermore, the chemical composition of commercial EO from USA is characterized by the abundance of *trans*-β-farnesene (42.59%) [100].

Chromatographic analysis of the chemical composition of *M. chamomilla* extracts revealed the presence of phenolic acids, flavonoids, coumarin, and amino acids. In addition, the extracts contain sterols, triterpens, saponins, tannins, and alkaloids [55,101]. The work of Elsemelawy indicated that Egyptian *M. chamomilla* powder was rich in flavonoids (luteolin O-acylhexoside and quercetin) and phenolic acids (ellagic acid, catechol, and chlorogenic acid) [102]. Another study on aqueous extract showed that the main compounds were also flavonoids (myricetin, quercetin, and naringenin) and phenolic acids (benzoic and rosmarinic acids) [53]. In their study on *M. chamomilla* from Slovakia, Petrul’ová-Poracká et al. reported for the first time the presence of three coumarins in methanolic extract, skimmin (umbelliferone-7-*O*-β-d-glucoside), daphnin (daphnetin-7-*O*-β-d-glucoside), and daphnetin (7,8-dihydroxycoumarin) [103]. On the other hand, coumarin, phenolic acid, and flavonoids contents of leaves methanolic extract were influenced by ethephon treatment [17]. In addition, the investigation of Kováčik et al. showed that the phenolic acid content of leaf rosettes was influenced by exposure to cadmium and copper. The mineral content of leaf rosettes and roots was also affected [18]. On the other hand, the results of the methanolic extracts from Iran *M. chamomilla* showed the highest amounts of luteolin and apigenin [10]. In addition, Sayyar et al. reported n-heptacosane (33.53%), a higher alkane, as the main component of ethanol extract [104]. The HPLC technique was used to determine the caffeoylquinic acid content of 34 samples from different geographical areas of China [105]. The findings showed the presence of six phenolic acids dominated by isochlorogenic acid A. The extract of *M. chamomilla* can also contain amino acids, such as proline and alanine [106]. *M. chamomilla* extract from Pakistan showed the dominance of amino acids proline and asparagine [16].

**Table 2 life-12-00479-t002:** Chemical composition of *Matricaria chamomilla* essential oils and extracts.

Country/Source	Part Used	Compounds Groups	Main Compounds	References
			Essential oils	
Morocco	Aerial parts	Terpenoids Coumarin	Chamazulene (26.11%), *cis*-β-farnesene (11.64%), eucalyptol (8.19%), *trans*-caryophyllene (5.95%), galaxolide (5.31%) Coumarin (6.01%)	[87]
Egypt	Flower	Terpenoids	α-Bisabolol oxide A (33–50.5%), *cis*-tonghaosu (10–18.7%), α-bisabolol oxide B (8.2–15.4%), α-bisabolone oxide A (5.4–14.6%), chamazulene (1.9–5.2%)	[13]
	Flower	Terpenoids	*Cis*-β-Farnesene (27%), D-limonene (15.25%), α-bisabolol oxide A (14.9%)	[88]
Turkey	Flower	Terpenoids	α-Bisabolol (27.36–38.6%), β-farnesene (25.05–30.15%), chamazulene (13.5–13.93%), Germancrene D (4.35–6.11%)	[89]
Iran	Flower	Terpenoids	α-Bisabolol oxide A (29.7–33.7%), chamazulene (18.76–20%), α-bisabolol oxide B (8.881–14.8%), α-bisabolone oxide A (6.64–8.3%), α-bisabolol (0.91–2.01%)	[93]
	Flower	Terpenoids	α-Bisabolol oxide A (37.2–44.5%), α-bisabolone oxide A (11.7–16.5%), *trans*-β-farnesene (13.3–15.4%), menthol (0–13%), *cis*-spiroether (5.6–9.9%), α-bisabolol oxide B (3–7.1%)	[15]
	Flower	Terpenoids	Bisabolol oxide A (7.31–51.31%), bisabolone oxide (8.35–39.97%), bisabolol oxide B (3.18–35.7%), *trans*-β-farnesene (2.05–19.68%), spathulenol (0–9.46%)	[11]
	Aerial part	Terpenoids	α-Bisabolol (17.51%), *cis*-*trans*-farnesol (8.72%), β-bisabolene (8.37%), *trans*-β-farnesene (5.48%), guaiazulene (4.36%), α-pinene (3.68%), limonene (3.24%)	[96]
	Flower	Terpenoids	Chamazulene (31.2%), 1,8-cineole (15.2%), β-pinene (10.1%), α-pinene (8.14%), α-bisabolol (7.45%), terpinen-4-ol (4.11%)	[95]
	Aerial part	Terpenoids	*Cis*-pinocamphone (not detected–73.54%) α-bisabolol oxide A (7.97–62.16%), chamazulene (1.67–15.08%), *trans*-β-farnesene (1.24–12.87%)	[14]
	Flower	Terpenoids	α-Bisabolone oxide A (45.64–65.41%), *trans*-γ-bisabolene (not detected—42.76%), *cis*-γ-bisabolene (not detected—40.08%), α-bisabolol oxide B (not detected—21.88%), chamazulene (not detected—19.22%)	[10]
	Flower	Terpenoids	*Trans*-β-farnesene (24.19%), guaiazulene (10.57%), α-bisabolol oxide A (10.21%), α-farnesene (8.7%), α-bisabolol (7.27%)	[94]
	Flower	Terpenoids	α-Bisabolone oxide A (11.9–63.5%), α-bisabolol oxide A (10.6–37.9%), α-bisabolol oxide B (2.4–23.9%)	[12]
Hungary	Flower	Terpenoids	α-Bisabolol oxide A (40.7%), chamazulene (14.3%), α-bisabolone oxide A (12.8%), α-bisabolol oxide B (8.7%)	[12]
Germany	Flower	Terpenoids	α-Bisabolol oxide A (39.1%), α-bisabolone oxide A (17.4%), α-bisabolol oxide B (17.1%), *cis*-Enyne-dicycloether (10.3%)	[12]
	Flower	Terpenoids	Bisabolol oxide A (54.1%), *cis*-enyne-dicycloether (19%), bisabolol oxide B (6.7%), bisabolone oxide A (4.5%)	[9]
Estonia	Flower	Terpenoids	Bisabolol oxide A (27.5–47.9%), bisabolone oxide A (1.6–17.1%), *cis*-enyne-dicycloether (11.7–14.9%), bisabolol oxide B (9.9–12.3%)	[9]
Greece	Flower	Terpenoids	Bisabolol oxide A (41.9%), *cis*-enyne-dicycloether (11.4%), bisabolol oxide B (6.4%)	[9]
Scotland	Flower	Terpenoids	Bisabolol oxide A (55.6%), *cis*-enyne-dicycloether (14%), bisabolol oxide B (8%), bisabolone oxide A (7.6%)	[9]
England	Flower	Terpenoids	Bisabolol oxide A (56%), *cis*-enyne-dicycloether (13.3%), bisabolol oxide B (7.1%), bisabolone oxide A (4.3%)	[9]
Latvia	Flower	Terpenoids	Bisabolol oxide A (51.9%), *cis*-enyne-dicycloether (13%), bisabolol oxide B (7.5%), *trans*-β-farnesene (5.3%)	[9]
Moldova	Flower	Terpenoids	α-Bisabolol (44.2%), *cis*-enyne-dicycloether (13.2%), bisabolone oxide A (12.4%), bisabolol oxide A (9.3%), bisabolol oxide B (6.6%)	[9]
Russia	Flower	Terpenoids	α-Bisabolol (23.9%), bisabolol oxide A (16.4%), *cis*-enyne-dicycloether (14.4%), bisabolol oxide B (10.5%), *trans*-Nerolidol (7.4%)	[9]
Czech	Flower	Terpenoids	α-Bisabolol (37%), *cis*-enyne-dicycloether (26.1%), chamazulene (9.8%), *trans*-β-farnesene (4.5%)	[9]
Armenia	Flower	Terpenoids	Bisabolol oxide B (27.2%), chamazulene (15.3%), bisabolol oxide A (12.6%), *cis*-enyne-dicycloether (12.6%), bisabolone oxide A (11.2%)	[9]
Ukraine	Flower	Terpenoids	Bisabolone oxide A (24.8%), α-bisabolol (17.1%), bisabolol oxide A (12.3%), bisabolol oxide B (11%), *cis*-enyne-dicycloether (8.8%)	[9]
Romania	Flower	Terpenoids	Bisabolol oxide A (70.2%), β-farnesene (6.47%), α-bisabolol oxide B (6.21%), *cis*-lanceol (5.071%)	[90]
Brazil	Flower	Terpenoids	α-Bisabolol oxide B (26.08%), β-farnesene (16.35%), bisabolol oxide A (14.7%), α-bisabolol (7.91%)	[91]
Italy	Aerial parts	Terpenoids	*Cis*-tonghaosu (11.8–45.9%), α-bisabolol oxide B (3.7–28.1%), α-bisabolol oxide A (2.7–19%), spathulenol (3.6–12.8%)	[92]
Australia	Flower	Terpenoids	Bisabolol oxide B (27.5%), bisabolol oxide A (27%), α-bisabolol (6.6%), *cis*-spiroether (6.1%), farnesene (4.5%), chamazulene (3.5%), *trans*-spiroether (0.6%)	[99]
USA	Commercial	Terpenoids	*Trans*-β-Farnesene (42.59%), bisabolol oxide A (21.2%), (E,E)-α-farnesene (8.32%), α-bisabolone oxide A (4.53%), α-bisabolol oxide B (4.43%), germacrene D (2.93%)	[100]
			Extracts	
Egypt	Flower and roots Powder	Flavonoids Phenolic acids	Luteolin O-acylhexoside (2801.99 ppm), quercetin (1765.01 ppm) Ellagic acid (1582.81 ppm), catechol (1104.49 ppm), chlorogenic acid (937.48 ppm)	[102]
	Flower Aqueous	Flavonoids Phenolic acids	Myricetin (1587.82 ppm), quercetin (927.72 ppm), naringenin (400.99 ppm) Benzoic acid (414.88 ppm), rosmarinic acid (370.59 ppm)	[53]
Slovakia	Leaf rosettes Methanol	Phenolic acids	Ferulic acid (196.8–512.5 μg/g), caffeic acid (66.8–106.1 μg/g), vanillic acid (45.6–71.1 μg/g), chlorogenic acid (12.6–26.2 μg/g), p-coumaric acid (14.4–26.1 μg/g)	[18]
	Flower or leaves Methanol	Coumarin	E-GMCA (9.82–17.8 mg/g), Z-GMCA (5.84–16.1 mg/g), herniarin (0.41–2.06 mg/g), daphnin (0.142–0.257 mg/g), skimmin (0.13–0.23 mg/g), umbelliferone (0.02–0.06 mg/g), daphnetin (trace-0.02 mg/g)	[103]
	Leaves Methanol	Coumarin Phenolic acids	E-GMCA (6.86–9.62 mg/g), Z-GMCA (1.22–6.6 mg/g) Vanillic acid (29.27–62.46 µg/g), caffeic acid (7.44–14.14 µg/g)	[17]
Iran	Flower Methanol	Flavonoids	Luteolin (2.2 mg/g), apigenin (1.19 mg/g)	[10]
	Not specified Ethanol	Alkane	n-Heptacosane (33.53%), 2,6,10,14,18,22-tetracosahexaene (16.71%), 1,2,2-trimethylcyclopropylamine (13.76%), 7-methoxy-2,3,4,5,6,7-hexahydro (6.13%), 1,2-benzenedicarboxylic acid (5.99%), Phenol, 4-(2-aminoethyl) (5.26%), hex-5-enylamine (4.48), 3-fluorophenethylamine (0.2%)	[104]
Pakistan	Not specified Aqueous	Amino acids	l-Proline (185 mg/mL), l-asparagine (97 mg/mL), aminobutyric acid (52 mg/mL), l-aspartic acid (45 mg/mL), l-alanine (43 mg/mL), l-glutamic acid (42 mg/mL)	[16]
China	Roots, stems, Leaves 70% aqueous methanol	Caffeoylquinic acids	Isochlorogenic acid A (0.1–5.15 mg/g), chlorogenic acid (0.03–4.08 mg/g), isochlorogenic acid C (0.06–3.17 mg/g), isochlorogenic acid B (0.03–2.45 mg/g), neochlorogenic acid (0.02–1.68 mg/g), cryptochlorogenic acid (0.005–0.33 mg/g)	[105]
	Flower Aqueous	Amino acids	Proline (4.24 mg/g), alanine (3.79 mg/g), isoleucine + leucine (2.59 mg/g), arginine + threonine (2.53 mg/g)	[106]

## 4. Pharmacological Interest

### 4.1. Antioxidant Activity

Several studies have investigated the antioxidant activity of *M. chamomilla* EO, extracts, and cell suspension culture (Table 3). These researches were carried out by several tests, including DPPH, ABTS, FRAP, β-carotene bleaching, ferrous ion chelating ability, and lipid peroxidation inhibition. The enzyme activities were also assessed for catalase, acetylcholine esterase, glutathione, peroxidase, ascorbate peroxidase, and superoxide dismutase.

DPPH assay is the most widely used to assess antioxidant activity. Using this test, *M. chamomilla* EOs showed the highest DPPH scavenging activity after 90 min of reaction [107]. On the other hand, *M. chamomilla* EOs had less activity than the used standards BHT and α-tocopherol in both DPPH and β-carotene bleaching tests, while it showed a comparable ferrous ion chelating ability than the citric acid standard [108]. The ABTS test was used to investigate the antioxidant power of EOs obtained by different extraction techniques (steam stripping, hydrodistillation, steam-dragging distillation with simultaneous steam extraction using an organic solvent, maceration, and supercritical fluid extraction) [109]. The results varied depending on the extraction technique, with the steam stripping technique providing the higher antioxidant activity in ABTS assay. This observation could be explained by the variation of EOs’ chemical composition depending on the extraction methods.

In another study on 13 cultivated *M. chamomilla*, the EOs and methanol extracts exhibited an interesting antioxidant activity using DPPH and FRAP assays [92]. This activity varied depending on the environmental factors and the chemical composition. Indeed, the highest activity was obtained by EOs rich in oxygenated compounds and extracts with high phenolic content. Comparing EOs and extracts, the authors reported higher DPPH activity from extracts, while both had similar activity using the FRAP test. Moreover, ABTS assay was used to study the antioxidant activity of two extracts (ethyl acetate extract and aqueous extract) from different plant parts (flower, leaf, stem, and root) [110]. The results showed that hydrophilic antioxidant activity (HAA) was significantly higher than the lipophilic antioxidant activity (LAA).

Abdoul-Latif et al., using the DPPH test, found that methanolic extract exhibited higher activity than EO, while EO showed a higher relative antioxidant activity in β-carotene-linoleic acid assay [111]. Other studies also showed the antioxidant activity of methanolic extract [46,112]. In addition, the activity varied depending on plant organ/tissue used, with separated parts exhibiting sometimes higher activity than whole herb [113]. On the other hand, Munir et al. [114] reported higher antioxidant activity with methanolic extract compared to ethanolic extract using DPPH assay, showing the role of extraction solvent in the antioxidant activity [114]. In the same way, Roby et al. found that methanol extract had the strongest DPPH scavenging activity, followed closely by ethanol, then diethyl ether and hexane extracts [115]. On the other hand, *M. chamomilla* ethanolic extract was also able to scavenge DPPH radicals [49,116]. Using also DPPH assay, Molnar et al. found that hydroethanolic extract obtained from processing waste fraction using maceration method exhibited the highest antioxidant activity compared to other extracts from processed chamomile flowers first class, unprocessed chamomile flowers first class, and pulvis [20]. This shows that extract obtained from waste can also exhibit interesting antioxidant capacity. In addition, Cvetanović et al. investigated the antioxidant activity of extracts obtained by several extraction techniques (microwave-assisted, Soxhlet, and ultrasound-assisted extraction) using two solvents (ethanol and water) compared to subcritical water extraction [117]. In general, subcritical water extract had the strongest DPPH scavenging capacity and ability to reduce Fe^3+^ to Fe^2+^ compared to other extracts. Furthermore, the temperature selected during subcritical water extraction had an influence on the antioxidant power of *M. chamomilla* aqueous extracts [118]. Extracts obtained at a temperature of 210 °C demonstrated the strongest DPPH activity, while extraction at 150 °C gave extracts with the highest ABTS and hydroxyl radical scavenging activities and lipid peroxidation inhibition. In the same way, Sotiropoulou et al. [119] investigated the effect of extraction temperature (25, 80, and 100 °C) on aqueous extracts’ DPPH capacity. The results showed that the extract at 80 °C had the highest activity, while no activity was reported by extract prepared at 25 °C. These results can be explained by the highest phenol content obtained at 80 °C compared to other temperatures. Indeed, high temperatures can allow the extraction of more polyphenols, but extremely high temperatures (100 °C) can lead to the loss of unstable ones. In addition, using a multivariate approach, Pereira et al. investigated the effect of several extraction parameters on the antioxidant capacity of hydroalcoholic extract obtained by dynamic maceration process. The results showed that extraction temperature, ratio of plant to solvent, and ethanol strength were the factors that exhibited most influence on the extract. Using the optimal conditions, the authors found that the extract was rich in flavonoids, apigenin, and apigenin-7-glycoside, and had high antioxidant activity close to the predicted results [120].

The enzymatic treatment of *M. chamomilla* aqueous infusion by hesperidinase and β-galactosidase led to a small increase in the percentage of DPPH radical-scavenging activity even though several phenolic compounds were altered by the treatment [121]. Using chicken liver tissue, Singh et al. found that *M. chamomilla* tea extract exhibited great antioxidant activity in lipid peroxidation tests [122]. In addition, the lipid peroxidation inhibition was influenced by treatment with electron beam irradiation [123]. Indeed, this treatment used for plant microbial decontamination caused a decrease of antioxidant activity in a dose-dependent way. This result was explained by the decrease of plant chemical compounds content, including flavonoid, tannins, and polyphenolcarboxylic acids. On the other hand, Hassanpour and Niknam found that the treatment of *M. chamomilla* cell suspension culture with a static magnetic field ameliorated its antioxidant activity and flavonoid metabolism [124]. Moreover, cells culture under clino-rotation induced antioxidant enzyme activity leading to growth and cell division [19]. Consequently, the parameters of cell culture also influence the antioxidant activity.

*M. chamomilla* EOs and extracts possess antioxidant activity that allows their use to prevent or treat diseases. In male Wistar rats, the intake of hydroalcoholic extract prevented the increase of superoxide dismutase globule and plasma malondialdehyde caused by a high cholesterol diet [125]. Additionally, in diabetic rats, the treatment with 10 and 20% *M. chamomilla* powder significantly decreased lipid peroxidation and increased catalase, acetylcholine esterase, and glutathione levels in serum [19].

**Table 3 life-12-00479-t003:** Studies/investigations on antioxidant activity of *M. chamomilla* essential oils, extracts, and other.

Part Used	Main Component	Experimental Method	Key Results	References
Essential oils				
Leaves	Not specified	DPPH β-carotene-linoleic acid assay	IC_50_ = 4.18 µg/mL Relative antioxidant activity = 12.69%	[111]
Leaves and flowers	Enyne-dicycloether (36.13–47.6%) Bisabolol oxide A (47.1%) β-Farnesene (30.2–37.62%)	ABTS	TEAC = 13.81–27.56 μmol TE/mL	[109]
Flower	α-Bisabolone oxide A (35.74%)	DPPH Ferrous ion chelating ability β-carotene bleaching	IC_50_ = 793.89 µg/mL IC_50_ = 1448.68 µg/mL 34.21%	[108]
Flower	*Trans*-β-Farnesene (29.8%)	DPPH	EC_50_ = 2.07 mg/mL	[107]
Aerial parts	*Cis*-tonghaosu (11.8–45.9%) α-Bisabolol oxide B (3.7–28.1%)	DPPH FRAP	TEAC = ~30–273.5 μmol TE/100 g DW TEAC = ~35–657.1 µmol TE/100 g DW	[92]
Extracts				
Aerial parts Methanol	Phenol content (~390–2689.2 mg GAE/100 g)	DPPH FRAP	TEAC = ~260–881.1 µmol TE/100 g DW TEAC = 137.2–1200.3 µmol TE/100 g DW	[92]
Flower Ethyl acetate Na phosphate buffer Leaf Ethyl acetate Na phosphate buffer Stem Ethyl acetate Na phosphate buffer Root Ethyl acetate Na phosphate buffer	Phenol content 11.29 mg GAE/g DW 21.78 mg GAE/g DW 9.21 mg GAE/g DW 10.56 mg GAE/g DW 8.13 mg GAE/g DW 13.45 mg GAE/g DW 4.16 mg GAE/g DW 4.41 mg GAE/g DW	ABTS	LAA = 3.51 mg TE/g DW HAA = 17.57 mg TE/g DW LAA = 1.47 mg TE/g DW HAA = 9.28 mg TE/g DW LAA = 1.49 mg TE/g DW HAA = 12.27 mg TE/g DW LAA = 1.48 mg TE/g DW HAA = 18.02 mg TE/g DW	[110]
Roots Methanol Ethanol	Not specified	DPPH	IC_50_ = 82.8% IC_50_ = 37.67%	[114]
Flower 70% ethanol Water	Phenol content (117.31–151.45 mg CAE/mL)	DPPH Reducing power	IC_50_ = 0.0211–0.0606 mg/mL EC_50_ = 0.578–0.922 mg/mL	[118]
Flower Water	Apigenin (231–1501 mg/kg) Luteolin-7-*O*-glucoside (166–1101 mg/kg)	ABTS DPPH Hydroxyl radical scavenging Lipid peroxidation inhibition	IC_50_ = 7.3–16.8 µg/mL IC_50_ = 10–45 µg/mL IC_50_ = 38.1–43.1 µg/mL IC_50_ = 28.7–35 µg/mL	[118]
Plant material Water	Not specified	DPPH	41.3–49.5%	[121]
Flowering plant Ethanol	Phenol content (284.6 ± 16 mg GAE/g DW)	DPPH	IC_50_ = 56.4 µg/mL	[49]
Flower Methanol Ethanol Diethyl ether Hexane	Phenol content (3.7 mg GAE/g DW) (3.5 mg GAE/g DW) (3.3 mg GAE/g DW) (2.4 mg GAE/g DW)	DPPH	EC_50_ = 0.0022 µmol EC_50_ = 0.0026 µmol EC_50_ = 0.0039 µmol EC_50_ = 0.0041 µmol	[116]
Aerial parts Ethanol (70%)	Phenol content (78.4 mg GAE/g DW)	DPPH	IC_50_ = 50 µg/mL	[116]
Leaves Methanol	Not specified	DPPH	IC_50_ = 65.8 μg/mL	[112]
Whole herb Stem Flower Methanol	Phenol content Whole herb (37.1 mg/kg DW) Stem (23.6 mg/kg DW) Flower (31.9 mg/kg DW)	DPPH	Whole herb: IC_50_ ~2.5 μg/mL Stem: IC_50_ ~2.4 μg/mL Flower: IC_50_ ~2.35 μg/mL	[113]
		Linoleic acid emulsion (30 h)	Whole herb: 63% Stem: 69% Flower: 60%	
		FRAP	Whole herb: absorbance ~1.8 Stem: absorbance ~0.88 Flower: absorbance ~0.9	
		Ferrous ions (Fe^2+^) chelating capacity	Whole herb: 73% Stem: 67% Flower: 85%	
		Superoxide radical scavenging activity	Whole herb: IC_50_ = 2.1 μg/mL Stem: IC_50_ = 2.8 μg/mL Flower: IC_50_ = 2.2 μg/mL	
Leaves Methanol	Not specified	DPPH β-carotene-linoleic acid assay	IC_50_ = 1.83 µg/mL Relative antioxidant activity = 11.37%	[111]
Flower Ethanol 50%	Umbelliferone content (11.80 mg/100 g) Herniarin content (82.79 mg/100 g)	DPPH	45.4–61.5%	[20]
Flowering parts Water	Phenol content (0.041- 0.165 mg GAE/mL)	DPPH	Inhibition = 2.53–4.62 µg TE/mL	[119]
Inflorescences Ethanol (74.7%)	Flavonoid content (4.11%)	DPPH	IC_50_ = 18.19 µg/mL	[120]
Flower Methanol (80%)	Phenol content (656.1 mg GAE/g FR)	DPPH FRAP	IC_50_ = 84.2 μg/mL IC_50_ = 13 mmol Fe^+2^/100 g	[46]
Flower Water	Phenol content (0.207 mg GAE/g)	Lipid peroxidation inhibition	Inhibition = 44.15%	[122]
Inflorescence Not specified	Flavonoid content (66.2–35.6 mg/g)	Lipid peroxidation inhibition	Inhibition = 10–100%	[123]
Whole Plant Ethanol 80%	Not specified	Superoxide dismutase Malondialdehyde	~2.2–3.1 U/mL plasma ~112–126 μmol	[125]
Flower and root Powder	Luteolin O-acylhexoside (2801.99 ppm)	Lipid peroxidation inhibition Catalase Acetylcholine esterase Glutathione	291.35–301.67 nmol 63.14–68.33 nmol 4.65–5.46 nmol 11.2–13.2 mg/g	[102]
Other				
Cell suspension culture	Phenol content (5.54–9.51 mg GAE/g DW)	DPPH Peroxidase Superoxide dismutase Ascorbate peroxidase	Inhibition = 55.1–76.72% ~2.75–3.75 unite/mg protein ~0.27–0.43 unite/mg protein ~2000–5000 unite/mg protein	[124]
Cell suspension culture	Total soluble sugar (63.71–96.04 mg/g FW)	Peroxidase Superoxide dismutase Catalase	~4.5–8 unite/mg FW ~0.5–0.75 unite/mg FW ~0.002–0.008 unite/mg FW	[19]

### 4.2. Antibacterial Activity

The antibacterial efficacy of *M. chamomilla* EOs and extracts was investigated by several studies (Table 4). Generally, the agar diffusion technique, using discs or wells, is the most used to screen the antibacterial activity of EOs and extracts. Using this technique, Stanojevic et al. [107] reported the antibacterial activity of *M. chamomilla* EO. The most sensitive strain was *Staphylococcus aureus*, while the most resistant one was *Pseudomonas aeruginosa*. Similarly, Owlia et al. [22] reported no activity against *P. aeruginosa* using the disc diffusion technique. However, the EO was able to reduce biofilm formation and alginate production, showing efficiency in controlling biofilm-producing bacteria. On the other hand, results of both diffusion and dilution techniques showed that *Bacillus subtilis* was the most sensitive bacteria to *M. chamomilla* EO from Morocco [87]. In addition, the EO showed the largest inhibition zone against *B. cereus* and the smallest MIC and MBC values against *S. aureus* [97]. Gram-positive bacteria also showed the smallest MIC values, as found by Silva et al. [126]. These results could be explained by differences in the cell wall structure since Gram-negative bacteria have a complex and rigid membrane rich in lipopolysaccharide, which limits the access of antimicrobial molecules [127]. In addition, the antibacterial activity of *M. chamomilla EO* was also reported against several *Streptococcus* species [128]. The commercial EO also showed antibacterial activity against several Gram-positive bacteria [129]. The results showed that the EO was more active against *S. aureus* MRSA compared to reference strains. On the other hand, *P. aeruginosa* was the most sensitive bacteria in both diffusion and micro-dilution assays [108,111], showing that Gram-negative bacteria could be more sensitive to EO compared to Gram-positive strains. In addition, Shakya et al. [130] found that *M. chamomilla* flower EO significantly reduced *Enterococcus faecalis* growth. More interestingly, Satyal et al. [131] reported that *M. chamomilla* EO exhibited the strongest activity against both *S. aureus* and *P. aeruginosa*. In addition, Hartmann and Onofre [132] reported the smallest MIC values against *Escherichia coli,* while no activity was found against *P. aeruginosa.* The largest inhibition zone was obtained against *S. aureus*, with the highest results when the EO was used without dilution (100%). Other factors can influence EO antibacterial activity, including plant origin, as found by Höferl et al. [27]. Indeed, the highest activity (MIC = 2000 µg/mL) was obtained against most bacteria by EO from South Africa (18.7% *trans*-β-farnesene) and Hungary (38.3% α-bisabolol).

Comparing the antibacterial activity of *M. chamomilla* EO and four extracts, Roby et al. [115] found that EO had the highest activity against all tested bacteria. The antibacterial activity increased with increasing concentrations of both EO and extracts. Among tested extracts, diethyl ether extract had the lowest activity, showing the effect of extraction solvent on antibacterial activity. Similarly, Ismail et al. [133] found that methanol and ethanol extracts studied at different concentrations were only active at the highest concentration and only against *S. aureus.* In addition, the aqueous extract was inactive against all strains at all tested concentrations. On the contrary, the aqueous extract was the strongest extract against most strains studied by Boudıeb et al. [134], except *Pseudomonas* sp., which was more sensitive to methanol extract. In their study on methanol, ethanol, petroleum ether, and ethyl acetate extracts, Abdalla and Abdelgadir [101] found that petroleum ether extract exhibited the highest activity (diameter-Φ between 22 and 26 mm) while ethyl acetate extract showed no activity against all tested bacteria. On the other hand, the study of Mariod et al. [135] on methanol, n-hexane, chloroform, ethyl acetate, and aqueous extracts from Sudanese and Egyptian *M. chamomilla* confirmed the effect of extraction solvent and plant origin on antibacterial activity. Indeed, Sudanese methanolic extract showed the highest activity, with better effect against *S. aureus* (W17) at concentrations of 50, 100, and 200 mg/mL. In another study, ethyl acetate extract exhibited a higher inhibitory effect against *Helicobacter pylori* compared to ethanol extract [136]. In addition, MBC values varied depending on ethanol percentage used (MBC = 125 µg/mL with ethanol 99.8%). On the other hand, ethanolic extract was only active against *P. aeruginosa*, while cyclohexane extract exhibited no antibacterial activity against all tested bacteria [137].

Several studies reported the antibacterial activity of *M. chamomilla* ethanolic extracts [117,138,139,140]. The results varied depending on the organ used [23,24,25]. Indeed, among 24 *S. aureus* MRSA studied, ethanolic extracts from flowers had an activity on 20 strains compared to 7 for leaf extracts [23]. The same results were found by Ahani Azari and Danesh [24] against MRSA strains. However, the authors found that ethanolic extract from leaves was the only one active against *P. aeruginosa* multidrug-resistant (16 strains). These findings were similar to the one found by Poudineh et al. [25]. On the other hand, methanol extracts also showed antibacterial activity [46,111,141,142]. The extract also exhibited an anti-adherence activity against all tested strains [26]. In addition, aqueous extracts have also demonstrated antibacterial activity [143,144] that varied depending on the plant organ used [145].

**Table 4 life-12-00479-t004:** Studies/investigations on in vitro antibacterial activity of *Matricaria chamomilla* essential oils and extracts.

Part Used	Main Component	Experimental Method	Tested Organism	Key Results	References
Essential oils					
Leaves	Not specified	Disc diffusion Micro-dilution	**Gram-positive** *Bacillus cereus* LMG 13569 *Listeria innocua* LMG 1135668 *Staphylococcus aureus* ATCC 9244 *Staphylococcus camorum* LMG 13567 *Streptococcus pyogenes* **Gram-negative** *Enterococcus faecalis* CIP 103907 *Escherichia coli* CIP 11609 *Salmonella enterica* CIP 105150 *Shigella dysenteriae* CIP 5451 *Proteus mirabilis* 104588 CIP *Pseudomonas aeruginosa*	Φ = 17 mm; MIC = 4; MBC = 4 µg/mL Φ = 20 mm; MIC = 2; MBC = 2 µg/mL Φ = 21 mm; MIC = 2; MBC = 2 µg/mL Φ = 22 mm; MIC = 2; MBC = 2 µg/mL Φ = 24 mm; MIC = 2; MBC = 2 µg/mL Φ = 14 mm; MIC = 4; MBC = 8 µg/mL Φ = 14 mm; MIC = 4; MBC = 8 µg/mL Φ = 20 mm; MIC = 2; MBC = 2 µg/mL Φ = 25 mm; MIC = 1; MBC = 1 µg/mL Φ = 17 mm; MIC = 4; MBC = 4 µg/mL Φ = 30 mm; MIC = 1; MBC = 1 µg/mL	[111]
**Aerial parts**	Chamazulene (26.11%)	Disc diffusion Micro-dilution	**Gram-positive** *Staphylococcus aureus* *Bacillus subtilis* **Gram-negative** *Escherichia coli* (ATB:57) B6N *Pseudomonas aeruginosa*	Φ = 14.13 mm; MIC = 8.33 µL/mL Φ = 15.2 mm; MIC = 6.25 µL/mL Φ = 13.27 mm; MIC = 8.33 µL/mL Φ = 13.07 mm; MIC = 8.33 µL/mL	[87]
**Flower**	Not specified	Disc diffusion (0.78–100%)	**Gram-positive** *Staphylococcus aureus* ATCC-25923 **Gram-negative** *Escherichia coli* ATCC-25922 *Pseudomonas aeruginosa* ATCC	Φ = 8.55–38.34 mm; MIC = 6.25% Φ = 9.31–12.32 mm; MIC = 1.56% No inhibition	[132]
Aerial parts (95% flowers)	*Trans*-β-farnesene (18.7–38.5%) α-Bisabolol (38.3%) α-Bisabolol oxide A (25%)	Macro-dilution	**Gram-positive** *Staphylococcus aureus* ATCC 6538 **Gram-negative** *Escherichia coli* ATCC 25922 *Salmonella abony* ATCC 6017 *Pseudomonas aeruginosa* ATCC 9027	MBC = 2000–8000 µg/mL MBC = 2000–8000 µg/mL MBC = 2000–8000 µg/mL MBC = 4000–8000 µg/mL	[27]
Aerial parts	α-Bisabolol oxide (38%)	Disc diffusion Micro-dilution	**Gram-positive** *Staphylococcus aureus* *Bacillus cereus* *Bacillus subtilis* **Gram-negative** *Shigella shiga* *Shigella sonnei* *Pseudomonas aeruginosa* *Proteus* sp.	Φ = 30 mm; MIC = 0.011; MBC = 0.13 µg/mL Φ = 36 mm; MIC = 0.022; MBC = 1.5 µg/mL Φ = 32 mm; MIC = 0.03; MBC = 1.5 µg/mL Φ = 25 mm; MIC = 0.14; MBC = 3 µg/mL Φ = 19 mm; MIC = 0.2; MBC = 3 µg/mL Φ = 19 mm; MIC = 4; MBC = 8 µg/mL Φ = 16 mm; MIC = 0.15; MBC = 3 µg/mL	[97]
Flower	α-Bisabolone oxide A (35.74%)	Disc diffusion Micro-dilution	**Gram-positive** *Staphylococcus aureus* ATCC 25923 *Enterococcus faecalis* ATCC 14506 **Gram-negative** *Escherichia coli* ATCC 25922 *Klebsiella pneumoniae* ATCC 13883 *Proteus vulgaris* ATCC 33420 *Pseudomonas aeruginosa* ATCC27853	Φ = 71.59%; MIC = 0. 25 mg/mL Φ = 106.7%; MIC = 0.12 mg/mL Φ = 99.66%; MIC = 0.17 mg/mL Φ = 75.04%; MIC = 0.15 mg/mL Φ = 89.15%; MIC = 0.21 mg/mL Φ = 108.77%; MIC = 0.04 mg/mL	[108]
Flower	Guaiazulene (25.6%)	(0.2–0.5 µg/mL) Disc diffusion Biofilm formation and adherence assay Quantitative assay of alginate	**Gram-negative** *Pseudomonas aeruginosa* 8821M	No inhibition Biofilm production = 0.17–0.64 µg/mL Alginate production = 190.33–549.33 µg/mL	[129]
Flower	Guaiazulene (25.6%)	Disc diffusion Macro-dilution	**Gram-positive** *Streptococcus pyogenes* PTCC 1447 *Streptococcus mutans* PTCC 1601 *Streptococcus salivarius* PTCC 1448 *Streptococcus faecalis* ATCC 29212 *Streptococcus sanguis* PTCC 1449	Φ = 9 mm; MIC = 0.1; MBC = 0.2 µg/mL Φ = 10 mm; MIC = 0.5; MBC = 1.5 µg/mL Φ = 9 mm; MIC = 0.5; MBC = 0.8 µg/mL Φ = 0.8 mm; MIC = 4; MBC = 7 µg/mL Φ = 8 mm; MIC = 0.5; MBC = 1 µg/mL	[129]
Commercial	Bisabolol and *trans*-β-farnesene	Macro-dilution	**Gram-positive** *Staphylococcus**aureus* MRSA (16 strains) *Staphylococcus* *aureus* (2 ATCC strains) *Staphylococcus* *epidermidis* ATCC 12228 *Enterococci* *faecalis* ATCC 51299 Vancomycin-resistant enterococci (9 strains)	MIC = 2–>4; MBC = 2–>4% MIC = MBC >4% MIC = MBC >4% MIC = MBC >4% MIC = MBC >4%	[129]
Aerial parts	*Trans*-β-farnesene (42.2%)	Micro-dilution	**Gram-positive** *Staphylococcus aureus* ATCC 29213 *Bacillus cereus* ATCC 14579 **Gram-negative** *Escherichia coli* ATCC 10798 *Pseudomonas aeruginosa* ATCC 27853	MIC = 313 μg/mL MIC = 625 μg/mL MIC = 625 μg/mL MIC = 313 μg/mL	[131]
Flower	Not specified	Disc diffusion Broth dilution Ex vivo	**Gram-negative** *Enterococcus* *faecalis*	Reduction = 2.91 CFU at day 14	[130]
Flower	*Trans*-β-Farnesene (29.8%)	Disc diffusion	**Gram-positive** *Staphylococcus aureus* WDCM 00032 *Listeria* *monocytogenes* WDCM 00020 *Salmonella enterica* WDCM 00030 **Gram-negative** *Escherichia coli* WDCM 00013 *Pseudomonas aeruginosa* WDCM 00024	Φ = 40 mm Φ = 13.33 mm Φ = 25 mm Φ = 31 mm No inhibition	[130]
Not specified	Chamazulene (31.48%)	Micro-dilution	**Gram-positive** *Staphylococcus aureus* (16 strains) **Gram-negative** *Escherichia coli* (16 strains)	MIC _90%_ = 2.9 mg/mL MIC _90%_ = 28.2 mg/mL	[126]
Flower	α-Bisabolol oxide A (48.22%)	Disc diffusion Micro-dilution	**Gram-positive** *Bacillus cereus* ATCC 11778 *Staphylococcus aureus* ATCC 13565 **Gram-negative** *Escherichia coli* O157 ATCC 1659 *Salmonella typhi* ATCC 13076	Φ ~12–22 mm; MIC = 10 µg/mL Φ ~12–26 mm; MIC = 10 µg/mL Φ ~7–19.5 mm; MIC = 12.5 µg/mL Φ ~10–21 mm; MIC = 12.5 µg/mL	[116]
**Extracts**					
Commercial Aerial parts Methanol Ethanol Petroleum ether	Not specified	Well diffusion	Gram-positive *Staphylococcus aureus* ATCC 25923 *Bacillus subtilis* NCTC 8236 Gram-negative *Escherichia coli* ATCC 25922 *Pseudomonas aeruginosa* ATCC 27853	Methanol: No inhibition Ethanol: Φ = 19 mm Petroleum ether: Φ = 25 mm Methanol: Φ = 17 mm Ethanol: Φ = 17 nm Petroleum ether: Φ = 26 mm Methanol: Φ = 17 mm Ethanol: Φ = 20 mm Petroleum ether: Φ = 23 mm Methanol: Φ = 17 mm Ethanol: Φ = 18 mm Petroleum ether: Φ = 22 mm	[101]
**Flower Ethanol**	Not specified	Broth microdilution	Gram-positive *Staphylococcus aureus* MRSA (30 strains)	MIC = 64–128 μg/mL	[140]
**Leaves Methanol**	Not specified	Disc diffusion Micro-dilution	Gram-positive *Bacillus cereus* LMG 13569 *Listeria innocua* LMG 1135668 *Staphylococcus aureus* ATCC 9244 *Staphylococcus camorum* LMG 13567 *Streptococcus pyogenes* *Enterococcus faecalis* CIP 103907 Gram-negative *Escherichia coli* CIP 11609 *Salmonella enteric* CIP 105150 *Shigella dysenteriae* CIP 5451 *Proteus mirabilis* 104588 CIP *Pseudomonas aeruginosa*	Φ = 17 mm; MIC = 100; MBC = 100 µg/mL Φ = 20 mm; MIC = 100; MBC ˃ 100 µg/mL Φ = 16 mm; MIC = 100; MBC = 100 µg/mL Φ = 19 mm; MIC = 100; MBC = 100 µg/mL Φ = 18 mm; MIC = 25; MBC = 50 µg/mL Φ = 13 mm; MIC = 100; MBC = 100 µg/mL Φ = 17 mm; MIC = 25; MBC = 25 µg/mL Φ = 17 mm; MIC = 100; MBC = 100 µg/mL Φ = 22 mm; MIC = 25; MBC = 25 µg/mL Φ = 15 mm; MIC = 50; MBC = 50 µg/mL Φ = 20 mm; MIC = 25; MBC = 25 µg/mL	[111]
Flower Ethanol	Not specified	Well diffusion (3.12–50 mg/mL) Micro-dilution	Gram-positive *Staphylococcus aureus* MRSA (14 strains) *Staphylococcus aureus* MRSA (6 strains) *Staphylococcus aureus* ATCC 29213 Gram-negative *Pseudomonas aeruginosa* ATCC 27,853 and multidrug-resistant (16 strains)	Φ = 10.3–12.7 mm at 25–50 mg/mL MIC = 6.25; MBC = 12.5 mg/mL Φ = 12.3 mm at 50 mg/mL MIC = 12.5; MBC = 25 mg/mL Φ = 12.1 mm at 50 mg/mL MIC = 12.5; MBC = 25 mg/mL No inhibition	[24]
Leaves Ethanol	Not specified	Well diffusion (3.12–50 mg/mL) Micro-dilution	Gram-positive *Staphylococcus aureus* MRSA (7 strains) *Staphylococcus aureus* ATCC 29213 Gram-negative *Pseudomonas aeruginosa* ATCC 27,853 and multidrug-resistant (16 strains)	Φ = 10.1 mm at 50 mg/mL MIC = 12.5; MBC = 25 mg/mL Φ = 9.8 mm at 50 mg/mL MIC–MBC > 50 mg/mL No zone; MIC = 12.5; MBC = 25 mg/mL	[24]
Flower Ethanol	Phenylindolizine (32.82%)	Well diffusion Micro-dilution	Gram-positive *Listeria monocytogenes* ATCC 19117 *Staphylococcus aureus* ATCC 25923 Gram-negative *Enterococcus faecalis* *Klebsiella pneumoniae* *Escherichia coli* ATCC 25922 *Enterobacter cloacae* *Acinetobacter baumannii*	Φ = 15 mm; MIC = 6.75 mg/mL Inhibition Inhibition Inhibition No inhibition No inhibition No inhibition	[139]
Leaves and flower Methanol Aqueous Chloroform	Phenol content 13.11 mg GAE/g DW 23.96 mg GAE/g DW 9.68 mg GAE/g DW	Disc diffusion	Gram-positive *Staphylococcus aureus* ATCC 6538 *Bacillus* sp. Gram-negative *Escherichia coli* ATCC 4157 *Pseudomonas* sp. ATCC 9027	Methanol: Φ = 6 mm Aqueous: Φ = 10 mm Chloroform: No inhibition Methanol: Φ = 9.66 mm Aqueous: Φ = 11.66 mm Chloroform: Φ = 9.33 mm Methanol: Φ = 6 mm Aqueous: Φ = 10.66 mm Chloroform: No inhibition Methanol: Φ = 22.5 mm Aqueous: Φ = 9 mm Chloroform: Φ = 10.33 mm	[134]
Flower Ethanol Cyclohexane	Not specified	Disc diffusion Broth dilution	Gram-positive *Staphylococcus aureus* ATCC 25923 Gram-negative *Pseudomonas aeruginosa* ATCC 27853 *Escherichia coli* ATCC 25922 *Salmonella* Typhimurium ATCC 14028	No inhibition Ethanol: Φ = 10 mm; MIC = 1000 mg/mL No inhibition No inhibition	[137]
Flower Ethanol	Phenol content (151.45 mg CAE/mL)	Micro-dilution	Gram-positive *Escherichia coli*	MIC = 39.1 µg/mL	[118]
Not specified Methanol	4-Amino- 1,5-pentandioic acid	Well diffusion (50 µL)	Gram-negative *Proteus mirabilis*	Φ = 6.01 mm	[141]
Not specified Methanol Ethanol Aqueous	Not specified	Well diffusion (250–1000 mg/mL)	Gram-positive *Staphylococcus aureus* Gram-negative *Escherichia coli* *Proteus* sp. *Klebsiella* sp.	Methanol: Φ = 12 mm at 1000 mg/mL Ethanol: Φ = 15 mm at 1000 mg/mL No inhibition No inhibition No inhibition	[133]
Flower Ethanol	Not specified	Well diffusion (10–100 μg/mL)	Gram-positive *Staphylococcus aureus*	Φ = 0–28 mm	[138]
Not specified Methanol	Phenol contents (1.24 mg GAE/g)	Disc-diffusion Macro-dilution	Gram-positive *Staphylococcus aureus* MTCC 7443 *Streptococcus mutans* MTCC 497 *Streptococcus mitis* MTCC 2695 *Streptococcus oralis* MTCC 2696 *Lactobacillus acidophilus* MTCC 10307 Gram-negative *Pseudomonas aeruginosa* MTCC 7453	Φ = 16.2 mm; MIC = 3.12 µg/mL Φ = 19.8 mm; MIC = 0.39 µg/mL Φ = 16.7 mm; MIC = 3.12 µg/mL Φ = 16.03 mm; MIC = 3.12 µg/mL Φ = 9.8 mm; MIC = 0.39 µg/mL No inhibition	[26]
Flower Ethanol (70, 96, 99.8%) Ethyl acetate	Not specified	Micro-dilution	Gram-negative *Helicobacter pylori* ATCC 43504	Ethanol: MIC = 62.5; MBC = 125–250 µg/mL Ethyl acetate: MIC = 31.3; MBC = 125 µg/mL	[136]
Not specified Methanol	Not specified	Well diffusion (12.5–200 mg/mL)	Gram-positive *Staphylococcus aureus* (2 strains) *Enterococcus faecalis* (3 strains) *Enterococcus durans* Sp. 33 Gram-negative *Proteus mirabilis* (3 strains) *Salmonella* S7 *Serratia* U11 *Providensia alcalifaciens* *Stenotrophomonas maltophilia*	Φ = 9–19 mm at 50–200 mg/mL Φ = 9.5–14 mm at 100–200 mg/mL Φ = 10–13 mm at 100–200 mg/mL Φ = 8–16 mm at 100–200 mg/mL Φ = 8–13 mm at 50–200 mg/mL Φ = 12 mm at 200 mg/mL Φ = 8–12 mm at 100–200 mg/mL Φ = 8 mm at 200 mg/mL	[135]
Flower Methanol	Phenol contents (656.1 mg CAE/g FR)	Diffusion (50 mg/mL) Micro-dilution	Gram-positive *Staphylococcus aureus* ATCC 6538 p *Streptococcus epidermidis* ATCC 12228 Gram-negative *Pseudomonas aeruginosa* ATCC 9027	Φ = 1.3 mm; MIC = 62.5 μg/mL Φ = 1 mm; MIC = 125 μg/mL Φ = 0.3 mm; MIC = 500 μg/mL	[46]
Aerial parts Aqueous	Not specified	Well diffusion (5–40 mg/mL)	Gram-positive *Staphylococcus aureus* Gram-negative *Escherichia coli*	Φ = 0.6–3.55 mm Φ = 0.6–3.6 mm	[143]
Stems Leaves Aqueous	Not specified	Disc diffusion	Gram-positive *Staphylococcus aureus* *Bacillus subtilis* Gram-negative *Escherichia coli* *Pseudomonas aeruginosa*	Stems: Φ = 22.7 mm Leaves: Φ = 21.8 mm Stems: Φ = 9.2 mm Leaves: Φ = 23.9 mm Stems: Φ = 9.9 mm Leaves: Φ = 23.7 mm Stems: Φ = 27.4 mm Leaves: Φ = 24.9 mm	[145]
Leaves Flower Ethanol	Not specified	Well diffusion Micro-dilution	Gram-negative *Pseudomonas aeruginosa* multidrug-resistant	Leaves: No zone MIC = 12.5; MBC = 25 mg/mL Flowers: No inhibition	[25]
Not specified Aqueous	Not specified	Disc diffusion (15–25%)	Gram-positive *Enterococcus faecalis* ATCC 24212	Φ = 20.62 mm at 25%	[144]
Flower Methanol Ethanol Hexane Diethyl ether	Phenol content (3.7 mg GAE/g) (3.5 mg GAE/g) (2.4 mg GAE/g) (3.3 mg GAE/g)	Disc diffusion (7.5–20 µg/disc) Micro-dilution	Gram-positive *Bacillus cereus* ATCC 11778 *Staphylococcus aureus* ATCC 13565 Gram-negative *Escherichia coli* O157 ATCC 1659 *Salmonella typhi* ATCC 13076	Methanol: Φ = 9–20 mm; MIC = 12.5 µg/mL Ethanol: Φ = 10–20 mm; MIC = 12.5 µg/mL Hexane: Φ = 9–21 mm; MIC = 12.5 µg/mL Diethyl ether: Φ = 7–18 mm; MIC=15 µg/mL Methanol: Φ = 11–19 mm; MIC=12.5 µg/mL Ethanol: Φ = 13–23 mm; MIC = 12.5 µg/mL Hexane: Φ = 10–23 mm; MIC = 10 µg/mL Diethyl ether: Φ = 8–19 mm; MIC = 15 µg/mL Methanol: Φ = 8–18 mm; MIC = 15 µg/mL Ethanol: Φ = 9–19 mm; MIC = 15 µg/mL Hexane: Φ = 8–19 mm; MIC = 15 µg/mL Diethyl ether: Φ = 7–15 mm; MIC = 17.5 µg/mL Methanol: Φ = 11–20 mm; MIC = 15 µg/mL Ethanol: Φ = 8–17 mm; MIC = 15 µg/mL Hexane: Φ = 8–19 mm; MIC = 15 µg/mL Diethyl ether: Φ = 6–16 mm; MIC = 15 µg/mL	[115]
Flower Ethanol	Not specified	Well diffusion (3.12–50 mg/mL) Micro-dilution	Gram-positive *Staphylococcus aureus* MRSA (14 strains) *Staphylococcus aureus* MRSA (6 strains) *Staphylococcus aureus* ATCC 29213	Φ = 10.3–12.7 mm at 25–50 mg/mL MIC = 6.25; MBC = 12.5 mg/mL Φ = 12.3 mm at 50 mg/mL MIC = 12.5; MBC = 25 mg/mL Φ = 12.1 mm at 50 mg/mL MIC = 12.5; MBC = 25 mg/mL	[23]
Leaves Ethanol	Not specified	Well diffusion (3.12–50 mg/mL) Micro-dilution	Gram-positive *Staphylococcus aureus* MRSA (7 strains) *Staphylococcus aureus* ATCC 29213	Φ = 10.1 mm at 50 mg/mL MIC = 12.5; MBC = 25 mg/mL Φ = 9.8 mm at 50 mg/mL MIC–MBC > 50 mg/mL	[23]
Leaves Methanol	Not specified	Well diffusion Micro-dilution	Gram-positive *Propionibacterium acnes* ATCC 11827 *Staphylococcus aureus* ATCC 6538P *Bacillus subtilis* MTCC 736 *Kocuria* sp KM 24375 Gram-negative *Escherichia coli* ATCC 8739 *Pseudomonas aeruginosa* ATCC 9027	Φ = 6 mm; MIC = 0.156 mg/mL No inhibition No inhibition No inhibition No inhibition No inhibition	[142]

### 4.3. Antifungal Activity

The antifungal activities of EOs and extracts obtained from different parts of *M. chamomilla* have been reported in the literature, suggesting great efficacy against a variety of fungal strains (Table 5). Most studies investigated the effect of *M. chamomilla* EO on *Candida* sp. Höferl et al. found that EOs from different origins were able to inhibit *C. albicans* growth with MFC = 2000 µg/mL, except EOs obtained from cultivated Indian plants dominated by α-bisabolol oxide A (25%), which showed MFC of 4000 µg/mL [27]. This reflects the influence of plant origin and EO chemical composition on antifungal activity. In addition, fluconazole-resistant and susceptible *C. albicans* strains isolated from Human Immunodeficiency Virus (HIV) positive patients with oropharyngeal candidiasis were inhibited by *M. chamomilla* EO, with a better effect on susceptible strains [146]. Other studies proved that *C. albicans* is more sensitive to EO than *Aspergillus* sp. [111,131]. Moreover, EL-Hefny et al. [88] found that EO antifungal activity was dose-dependent, with the best results against *A. niger*. Similar to their results on antibacterial activity, Mekonnen et al. reported no effect of *M. chamomilla* EO from Ethiopia on two *Aspergillus* sp. and two *Trichophyton* sp. Strains [147].

Comparing the antifungal capacity of *M. chamomilla* EO and extracts, Abdoul-Latif et al. [111] found that EO had better activity than methanolic extract. On the other hand, Roby et al. found that EOs and different extracts (methanol, ethanol, diethyl ether, and hexane) had a dose-dependent activity [115]. Among chloroform, methanol, and aqueous extracts, only chloroform showed antifungal activity (Φ = 6 mm) against *C. albicans* and *Fusarium* sp. [134]. This shows the effect of extracts on antifungal activity. In another study, Hameed et al. found that methanolic plant extract had an activity on *A. terreus* [141], while Lavanya et al. reported no activity against four *Candida* sp. (*C. albicans*, *C. tropicalis*, *C. parapsilosis*, and *C. krusei*) [26]. On the other hand, ethanolic flower extract also showed an antifungal activity [117]. Moreover, hydroalcoholic extract of *M. chamomilla* caused a significant decrease in *Saccharomyces cerevisiae* growth and cell survival [148]. In addition, aqueous extracts also showed antifungal activity [143,144]. On the other hand, seed aqueous extracts obtained at different pH (acidic, neutral, and alkaline) exhibited the same antifungal activity against *A. niger* and *P. citrinum* [149]. In addition, their sulfated derivatives exhibited a close antifungal activity even though they had higher phenolic content. In another study, Seyedjavadi et al. [150] isolated a novel peptide (AMP1) from *M. chamomilla,* with antifungal activity against *C. albicans* and *Aspergillus* sp. This shows that *M. chamomilla* can be source of interesting antifungal molecules.

**Table 5 life-12-00479-t005:** Studies/investigations on in vitro antifungal activity of *Matricaria chamomilla* essential oils, extracts, and other.

Part Used	Main Component	Experimental Method	Tested Organism	Key Results	References
Essential Oil					
Leaves	Not specified	Disc diffusion Micro-dilution	*Candida albicans* ATCC 10231 *Candida albicans* *Aspergillus* *niger* *Aspergillus* sp.	Φ = 20 mm; MIC = MFC = 1 µg/mL Φ = 19 mm; MIC = MFC = 2 µg/mL Φ = 17 mm; MIC = MFC = 2 µg/mL Φ = 14 mm; MIC = 16; MFC > 16 µg/mL	[111]
Flower	*Cis*-β-farnesene (27%)	Agar dilution (25–100 µL/mL)	*Aspergillus flavus* AFl375 *Aspergillus niger* FC24771 *Aspergillus terreus* Y.H. Yeh V0103 *Fusarium culmorum* CBS 128537	Φ = 10.66–52.33% Φ = 89.66–100% Φ = 87–84% Φ = 91–86.66%	[88]
Aerial parts (95% flowers)	*Trans*-β-farnesene (18.7–38.5%) α-Bisabolol (38.3%) α-Bisabolol oxide A (25%)	Macro-dilution	*Candida albicans* ATCC 10231	MFC = 2000–4000 µg/mL	[27]
Flower	α-Bisabolol oxide A (48.22%)	Disc diffusion (7.5–20 µg/disc) Micro-dilution	*Candida albicans* ATCC 10231 *Aspergillus flavus* ATCC 16875	Φ ~14–26 mm; MIC = 19 µg/mL Φ ~9–23 mm; MIC = 12.5 µg/mL	[115]
Aerial parts	*Trans*-β-farnesene (42.2%)	Micro-dilution	*Candida albicans* ATCC 10231 *Aspergillus niger* ATCC 16888	MIC = 313 μg/mL MIC = 625 μg/mL	[131]
Flower	α-Pinene (22.10%)	Broth dilution	*Candida albicans* (30 resistant) *Candida albicans* (30 susceptible)	MIC = 1700; MFC = 2300 μg/mL MIC = 1550; MFC = 2200 μg/mL	[146]
Extracts					
Leaves Methanol	Not specified	Disc diffusion Micro-dilution	*Candida albicans* ATCC 10231 *Candida albicans* *Aspergillus niger* *Aspergillus* sp.	Φ = 15 mm; MIC = MFC = 100 µg/mL Φ = 15 mm; MIC = MFC = 100 µg/mL Φ = 14 mm; MIC = 100; MFC > 100 µg/mL Φ = 13 mm; MIC = 200; MFC > 200 µg/mL	[111]
Leaves and flower Methanol Aqueous Chloroform	Phenols	Disc diffusion	*Candida albicans* ATCC 24433 *Fusarium* sp.	Chloroform: Φ = 6 mm Other extracts: No inhibition Chloroform: Φ = 6 mm Other extracts: No inhibition	[134]
Flower Ethanol	Phenol content (151.45 mg CAE/mL)	Micro-dilution	*Aspergillus niger*	MIC = 39.1 µg/mL	[117]
Not specified Methanol	4-Amino- 1,5-pentandioic acid	Well diffusion	*Aspergillus terreus*	Φ = 5.89 mm	[141]
Flower Alcohol 70%	Not specified	Spectrophotometer device	*Saccharomyces cerevisiae*	Growth decrease = 48% at 3000 μg/mL	[148]
Aerial part Aqueous	Not specified	Well diffusion	*Candida albicans*	Φ = 0.26–2.56 mm; MIC = 5–40%	[143]
Seeds Aqueous Sulfated derivatives	Phenol content (16.4–19.7 mg GAE/g) (19.2–22.4 mg GAE/g)	Disc diffusion	*Penicillium citrinum* *Aspergillus niger*	Aqueous: Φ = 10–12 mm Sulfated derivatives: Φ = 10–12 mm Aqueous: Φ = 10 mm Sulfated derivatives: Φ = 7–10 mm	[149]
Not specified Aqueous	Not specified	Disc diffusion (15–25%)	*Candida albicans* ATCC 24433	Φ = 24.16 mm at concentration of 25%	[144]
Flower Methanol Ethanol Hexane Diethyl ether	Not specified	Disc diffusion (7.5–20 µg/disc) Micro-dilution	*Candida albicans* ATCC 10231 *Aspergillus flavus* ATCC 16875	Methanol: Φ = 15–23 mm; MIC = 10 µg/mL Ethanol: Φ = 8–21 mm; MIC = 12.5 µg/mL Hexane: Φ = 9–23 mm; MIC = 10 µg/mL Diethyl ether: Φ = 8–20 mm; MIC = 15 µg/mL Methanol: Φ = 18–24 mm; MIC = 12.5 µg/mL Ethanol: Φ = 11–18 mm; MIC = 15 µg/mL Hexane: Φ = 8–20 mm; MIC = 12.5 µg/mL Diethyl ether: Φ = 6–21 mm; MIC = 17.5µg/mL	[115]
Other					
Flower	Peptide AMP1	Broth microdilution	*Candida albicans* *Aspergillus* sp.	MIC = 3.33–6.66 μmol MIC = 6.66–13.32 μmol	[150]

### 4.4. Antiparasitic and Insecticidal Activities

Several studies have investigated the capacity of *M. chamomilla* EOs and extracts to inhibit the growth of a wide range of parasites and insects. An in vitro evaluation of the leishmanicidal activity of Tunisian *M. chamomilla* EOs was carried out [151]. The results showed that EOs exhibited a good activity on the promastigotes (an extracellular and motile form) of *Leishmania amazonensis* (IC_50_ = 10.8 μg/mL after 96 h) and *L. infantum* (IC_50_ = 10.4 μg/mL after 96 h), while α-bisabolol was able to activate programmed cell death effects in the promastigote. In another study by Hajaji et al., the activity of α-bisabolol against *Acanthamoeba castellani* has been investigated. The results showed that α-bisabolol has amoebicidal activity with IC_50_ = 20.83 µg/mL and IC_90_ = 46.60 µg/mL, and it was able to increase the plasmatic membrane permeability and to decrease ATP levels [152].

*M. chamomilla* EOs from Nepal was screened for larvicidal activity against glassworm (*Chaoborus plumicornis*), insecticidal activity against fruit fly (*Drosophila melanogaster*), nematicidal activity against *Caenorhabditis elegans* and *Artemia salina* [131]. The results showed no notable toxicity on these organisms. Nevertheless, *M. chamomilla* EOs was found to be insecticidal against the desert locust Schistocerca gregaria 3rd nymphal instar (LD_50_ = 1.59 μg/g body weight) [153] and the saw-toothed grain beetle *Oryzaephilus surinamensis* (LC_50_ of 0.59% after 3 days) [154]. *M. chamomilla* EOs showed potent acaricidal activity against the red spider mite *Tetranychus urticae* (LC_50_ = 0.65%) [155] but was less active against the cattle fever tick *Rhipicephalus annulatus* (LC_50_ > 8%) [156]. In another research, *M. chamomilla* EOs rich in α-bisabolol oxide A showed nematicidal activity against the parasitic Anasakis L3 with 100% mortality at a concentration of 125 μg/mL [157]. Also, Höferl et al. tested the insecticidal activity of six EOs from *M. chamomilla* against larvae and adult mosquitoes of *Aedes aegypti*. The results varied depending on plant origin, with EOs from South Africa exhibiting the highest activity (LD_50_ = 2.9 ppm). These findings were related to EOs high content in steroidal spiroethers (12%) [27].

The anti-Acanthamoeba activity of flower extracts of Tunisian *M. chamomilla* was evaluated on Acanthamoeba castellanii [158]. The methanolic extract has shown a potent anti-acanthamoeba activity (IC_50_ = 66.23 μg/mL) which is attributed to a coumarin mixture. The anti-helminthic activity of the extracts from *M. chamomilla* flowers was evaluated on the egg and adult stages of Haemonchus contortus, which is a gastrointestinal parasite of small ruminants [159]. Methanolic and aqueous extracts showed higher inhibitory effects on egg hatching (IC_50_ of 1.52 and 2.55 mg/mL, respectively). After 8 h, methanolic extract induced 91.77% mortality at the highest concentration tested (8 mg/mL), while the aqueous extract induced only 75.05% mortality at the same concentration. Another study showed that methanol extract (at 1024 μg/mL) was the most active as an anti-helminthic against Haemonchus contortus [159]. The percentage of ovicidal activity was 37.5% for the egg hatch test, and the percentage of larvicidal activity was 84% for the larval development test. Concerning the mosquitocidal activity, Al-Mekhlafi et al.have tested the larvicidal and ovicidal effects of the combination of *M. chamomilla* and Foeniculum vulgare hexane extracts against Culex pipiens. The mixture obtained showed a larvicidal activity with LC_50_ of 100.3 mg/mL after 72 h exposure [28]. In addition, an ovicidal activity was reported with a decrease in egg hatchability from 95 to 15% at doses ranging from 62.5 to 500 mg/mL. The larval mortality ranged from 13.33 to 93.33% at doses ranging from 31.25 to 250 mg/mL. In another study, *M. chamomilla* ethyl acetate extract showed the most promising larvicidal activity against Culex pipiens, with 90% mortality at concentration 358.9 μg/mL after 48 h of exposure [160]. Treatment of eggs with concentration of 240 μg/mL showed 86.49% hatchability, and the life cycle could not be completed because all the larvae were dead (100% mortality).

### 4.5. Antidiabetic Activity

The activity of *M. chamomilla* extract and isolated apigenin, apigenin-7-*O*-glucoside, cis and trans-2-hydroxy-4-methoxycinnamic acid glucosides against α-amylase and maltase have been tested [29,161]. The results showed that the extract and the compounds exhibited a concentration-dependent inhibition on both enzyme activities. The highest α-amylase and maltase inhibition was obtained by apigenin and apigenin-7-*O*-glucoside, respectively. Furthermore, these two flavonoids were able to restrict sucrose and glucose transports and regulate sugar absorption. Moreover, another study reported that *M. chamomilla* hydro-methanolic extract and some isolated compounds inhibited rat lens aldose reductase activity [162]. In addition, 3,5-*O*-di-caffeoylquinic acid and luteolin-7-*O*-β-d-glucuronide suppressed sorbitol accumulation in rat lens under high-glucose conditions, while luteolin-7-*O*-β-d-glucuronide and luteolin suppressed advanced glycation end products formation. Furthermore, *M. chamomilla* ethanolic extract demonstrated anti-glycation properties with IC_50_ of 264.2 µg/mL for lipase inhibition activity [163].

### 4.6. Anti-Tumoral Activity

*M. chamomilla* extracts and EOs have also been studied for their anti-tumoral properties on several cancer cell lines. The anticancer activity of *M. chamomilla* EO was evaluated on human breast carcinoma (MCF-7) cell line by Ali [30]. The results showed that EOs inhibited the cell proliferation in a dose-dependent manner, with 89% inhibition after 24 h exposure at the highest concentration, 640 μg/mL. On the other hand, EO anticancer activity against two species of human promyelocytic leukemia cell lines (HL-60 and NB4) was tested [164]. The EOs were able to inhibit both cell lines growth, with higher dead percentages against NB4 cells (86.03% at 200 µg/mL) compared to HL-60 cells (78.4% at 200 µg/mL). In addition, *M. chamomilla* hydroalcoholic extracts from aerial parts or roots revealed an anti-proliferative effect on human breast cancer cells [31,165]. The IC_50_ was 785 g/mL against MDA-MB-468 and 921 g/mL against MCF-7 for aerial parts extracts and 1560 g/mL against MCF-7 for root extracts. The methanolic extract of *M. chamomilla* has been tested by Fraihat et al. [166] on two solid human melanocyte tumor cell lines, A375.S2 and WM1361A. In this study, results showed an inhibition only in the proliferation of the melanotic WM1361A cell line (IC_50_ = 25.2 g/mL). On the other hand, Cvetanović et al. [117] found that the extraction method impacted the anticancer efficacy of *M. chamomilla* extracts. Indeed, subcritical water extracts revealed the most effective cytotoxic activity against murine fibroblast cell line (IC_50_ = 19.65 μg/mL), human cervical carcinoma cell line Hep2c (IC_50_ = 20.54 μg/mL), and human rhabdomyosarcoma cell line (IC_50_ = 30.54 μg/mL). Antitumor potentials of water extracts of *M. chamomilla* seeds obtained at different pH conditions and their corresponding sulfated derivatives against the Ehrlich ascites carcinoma cells were evaluated by [167]. All extracts slightly inhibited the growth of the Ehrlich ascites carcinoma cell line at 3 different concentrations (300, 600, and 900 μg/mL). The anticancer properties of *M. chamomilla* appear to be linked to apoptosis and necrosis, as well as to a decrease in migration and invasion capacities of oncogenic cells [31,167].

### 4.7. Anti-Inflammatory Activity

The anti-inflammatory effect of *M. chamomilla* extracts has been reported [32]. According to the findings of this study, the anti-inflammatory activity of *M. chamomilla* ethanolic extract on macrophages was associated with a decrease in nitric oxide production and cell viability, while on lymphocytes, it was related to the induction of anti-inflammatory cytokine production (IL-10) and the decrease in cell viability. On the other hand, *M. chamomilla* aqueous extract caused a reduction of nitric oxide production and an increase in cell viability in macrophages, while it was an effective T helper Th2 suppressor by disrupting the Th1/Th2 balance. The difference between these two extracts could be attributed to the presence of different active compounds. In another study, Singh et al. investigated the anti-inflammatory properties of *M. chamomilla* tea extract. The results showed that extract caused inhibition of protein denaturation and stabilization of human red blood cell membrane, indicating its anti-inflammatory properties [122].

The anti-inflammatory activity of *M. chamomilla* was also investigated in animal models. According to Wu et al., the volatile and non-volatile components of M. chamomilla, essential oil, flower water, and aqueous extract, can all significantly inhibit swelling of mouse ears caused by xylene, pedal swelling caused by carrageenan in rats, and the increase of celiac capillary vessel permeability in mice. They also had a significant inhibitory effect on the increase in prostaglandin E2 and nitric oxide levels in rat pedal edema caused by carrageenan [168]. Furthermore, the effects of *M. chamomilla* hydroalcoholic extract on the level of inflammatory blood indicators were investigated on rats by Nargesi et al. [125]. Treatment with 110 mg/kg hydroalcoholic extract prevented a significant increase in serum levels of Tumor Necrosis Factor-α (TNF-α), C-Reactive Protein (CRP), Interleukin 6 (IL-6), and fibrinogen. On the other hand, the combination of ethanolic extract and diclofenac or indomethacin, two non-steroidal anti-inflammatory drugs, showed interesting synergic anti-inflammatory effects on carrageenan-induced paw inflammation and stomach damage in rats [169].

## 5. Phyotherapeutical Applications

Traditionally, *M. chamomilla* has been used to treat several diseases. Nowadays, studies have demonstrated the therapeutic potential of this plant in animal and human studies. Indeed, *M. chamomilla* showed an interesting effect on the nervous system of rats by improving learning, memory [104,116,170,171], and motor function [172]. In mice, Can et al. [173] found that EOs had a stimulant effect on the central nervous system similar to that of caffeine. In a clinical study, the oral administration of *M. chamomilla* extract caused a sedative effect on elderly people, improving their sleep quality [174]. In another study, EOs reduced crying and fussing in breastfed colicky infants [175]. In addition, *M. chamomilla* EOs can exhibit sedative effects against withdrawal syndrome in narcotics anonymous [176]. On the other hand, clinical studies showed that *M. chamomilla* could be used to treat anxiety and depression [177,178], including anxiety before esophagogastroduodenoscopy [179].

*M. chamomilla* tea was effective in reducing pain in patients after orthopedic surgery [45]. In addition, dermal application of flower EOs by patients with knee osteoarthritis decreased their need for analgesic acetaminophen and ameliorated physical function and stiffness [180]. In another clinical trial, Zargaran et al. found that *M. chamomilla* oleogel can be used to relieve pain in patients with migraine without aura [181]. Additionally, clinical studies demonstrated that *M. chamomilla* could exhibit an interesting analgesic effect on women during childbirth [182,183]. The plant was also efficient in relieving the pain of mild to moderate mastalgia, breast pain often preceding the menstrual period [184]. In addition, some reviews gathered studies on the use of *M. chamomilla* in the treatment of premenstrual syndrome [185], primary dysmenorrhea, and reducing menstrual bleeding [186]. On the other hand, *M. chamomilla* extracts can exhibit an effect on male and female reproductive systems of rats by influencing sexual hormones level [34,187]. Moreover, the extract showed a protective effect against formaldehyde in male rats’ reproductive system [49] and against torsion/detorsion-induced damages on adult rat testis tissue [188] and ovary tissue [189]. On the other hand, *M. chamomilla* extracts showed a therapeutic effect against thyroid damage [190] and kidney dysfunction [191] associated with polycystic ovary syndrome in female rats.

Other studies showed the protective effect of *M. chamomilla* on kidney and liver in animal models [47,192,193]. In animal models, several studies showed that *M. chamomilla* could be used to treat diabetes [102,194,195,196,197]. In a clinical trial on patients with type 2 diabetes, Rafraf et al. [35] found that short-term intake of *M. chamomilla* tea can control fatty acids and blood sugar levels and increase insulin sensitivity. In their review, Bayliak et al. [36] reported the possible use of *M. chamomilla* to treat obesity and related metabolic disorders. Moreover, in both animal and clinical studies, Awaad et al. [37] found that *M. chamomilla* had anti-hypertensive activity, decreasing the risk for various cardiovascular diseases.

In addition to anti-cancer activity reported before, *M. chamomilla* can also be used as a chemo-preventive agent [198]. Indeed, the findings showed that aqueous extract had a protective effect against 1,2-dimethylhydrazine that induced colorectal cancer in mice. Moreover, *M. chamomilla* showed antinociceptive effects against vincristine [199] and formalin [200] in animal models, showing its possible use to treat or attenuate negative side effects of chemotherapy. Clinical studies also showed the ability of *M. chamomilla* to reduce nausea [201], anxiety, and depression [202] in patients undergoing chemotherapy.

In rats, *M. chamomilla* exhibited therapeutic gastrointestinal effects on diarrhea [40] and gastric ulcer [39]. In addition, *M. chamomilla* showed a gastroprotective effect against alcohol-induced ulcer injury in rat gastric mucosa [48]. On the other hand, a traditional Brazilian herbal medicine (Arthur de Carvalho Drops^®^), prepared from plants extracts including M. chamomilla, showed beneficial effects for the treatment of gastrointestinal disorders in rats [38]. In a randomized controlled trial, Khadem et al. [203] found that the topical application of *M. chamomilla* EOs on the abdominal region of patients after cesarean section ameliorated their postoperative bowel activity.

In their study, Park et al. [204] reported that ethanol extract of *M. chamomilla* was efficient in treating muscle wasting in mice with dexamethasone-induced muscle atrophy. On the other hand, *M. chamomilla* methanol extract showed anti-allergic activity against compound 48/80 by reducing scratching behavior in mice [41]. This result was explained by the extract’s ability to inhibit histamine release from mast cells. In addition, *M. chamomilla* extract showed potential to heal wound [46] and atopic dermatitis-like lesions [205] in animal models. Among *Matricaria genus* is one of the most used in treating skin diseases. A number of patents and medicines have been developed using *M. chamomilla* EOs and extracts to treat skin diseases [42]. In addition, it has been used for the preparation of skincare formulations [206]. On the other hand, Jiménez Delgado et al. [207] found that *M. chamomilla* infusion can help reduce the dark rings under the eyes and periocular zone swelling. A commercial eye drop (Dacriovis™), containing extracts from *M. chamomilla* and *Euphrasia officinalis*, was found to exhibit a protective effect on human corneal epithelial cells from Ultraviolet B exposure [43]. Indeed, the eye drop showed antioxidant and anti-inflammatory activities, allowing it to provide protection against cell death and ameliorate wound healing. On the other hand, *M. chamomilla* can also be used to treat a number of oral diseases. As a saliva substitute, this plant was clinically used against burning mouth syndrome [44] and xerostomia (dry mouth sensation) [208]. As a mouthwash, *M. chamomilla* was used to treat gingivitis, allowing the decrease of biofilm accumulation and gingival bleeding [209]. In addition, Braga et al. [210] found that a mouth rinse containing *M. chamomilla* aqueous extract showed an anti-caries effect. An orabase containing chamomile extract relieved pain in patients with oral mucosal minor aphthous stomatitis [211].

## 6. Other Applications

The possible use of *M. chamomilla* as an anesthetic agent in aquaculture was reported [50,212]. In addition, the plant has been used by farmers as supplementary animal feeds. In rabbits, Alsaadi et al. [51] reported that aqueous flower extract promoted animal growth and had a positive effect on biochemical and hematological parameters. In another study, the use of *M. chamomilla* as a feed supplement positively influenced the intake of *Juniperus phoenicea* by goats [213]. *M. chamomilla* extracts have also been investigated as a food preservative in cottage cheese [59,214], yogurts [63], and biscuits [52]. On the other hand, the antifungal potential of *M. chamomilla* allows its use as an agricultural tool. In the greenhouse, Ghoniem et al. [53] reported the possible use of *M. chamomilla* aqueous extract to control *Pythium ultimum* fungus in bean crops. In addition, the possible use of *M. chamomilla* as a natural surfactant was studied. Shadizadeh and Kharrat [54] found that hydroglycolic extract can be used as a surfactant for a chemical enhanced oil recovery process since it decreased the oil-water interfacial tension. In another study, Ugi et al. [55] used *M. chamomilla* as an environmentally friendly inhibitor for the management of water corrosion of federated mild steel.

Although EOs and extracts have several biological activities, their application in industrial fields is limited by their low stability, low solubility, and high evaporation. The encapsulation allows the protection and target delivery and can also enhance biological activities [215]. Some studies incorporated *M. chamomilla* EOs and extracts into nanoparticles in order to improve their pharmacological properties. Indeed, Das et al. [56] prepared a Pickering emulsion of *M. chamomilla* EOs stabilized with modified Stöber silica nanoparticles. The Pickering nanoemulsions showed higher antibacterial and antifungal activities than that of emulsion stabilized with Tween 80 and ethanolic solution. The nanoparticles acted as a stabilizer, allowing the controlled release of EOs from the emulsion system. On the other hand, *M. chamomilla* extracts incorporated in silver nanoparticles demonstrated higher antibacterial and antifungal activities, explained by the synergistic effect between nanoparticles and extract, high localized concentration of extract, and size-specific nanoparticle efficacy [62]. In addition, Negahdary et al. [216] reported good activity of *M. chamomilla* silver nanoparticles against *S. aureus* growth and *C. albicans* biofilm. In another study, silver nanoparticles also exhibited higher activity on bacteria from dairy products [57]. In addition, silver nanoparticles prepared with *M. chamomilla* aqueous extract exhibited anticancer activity against human lung adenocarcinoma cell line (A549) [60]. On the other hand, silver nanoparticles containing aqueous extract exhibited catalytic activity against Rhodamine B under UV irradiation and thus can be considered a promising solution for wastewater treatment [61]. In their study, Karam et al. [58] found that chitosan nanocapsules containing *M. chamomilla* EOs had activity against *Leishmania amazonensis* , allowing its use to treat leishmaniasis. In addition, EO nanocapsules showed a significant reduction in cytotoxicity against mammalian cells compared to free EOs. In another study, *M. chamomilla* aqueous extract microencapsulated in alginate exhibited higher antioxidant activity when incorporated into cottage cheese [59]. However, other reviews on *M. chamomilla* highlighting several aspects of great interest can be consulted [217,218].

## 7. Conclusions

In this current review about M. chamomilla, we reported taxonomy and synonym, botanical and ecology description, geographic distribution, ethnomedicinal use, phytochemistry, pharmacological properties, medicinal and other applications, and encapsulation solutions. Traditionally, *M. chamomilla* was used to treat a variety of diseases, including diabetes, nervous disorders, diarrhea, angina, canker sore, abscess, microbial infections, painful menstruation, antiseptic, anti-inflammatory, sciatic pain, throat, ear, and skin, and stomach disorder. Moreover, antioxidant, antibacterial, antifungal, anticancer, antidiabetic, antiparasitic, antipyretic, anti-inflammatory, anti-osteoporosis, and analgesic activities of *M. chamomilla* EOs and extracts have been identified in in vitro and in vivo studies. The chemical composition of this plant from different countries of the world also has been reported in this work. Almost all studies have concentrated on the plant’s flower components. The abundance of terpenoids present in EOs and phenolic compounds present in extracts of *M. chamomilla* has been shown through the phytochemical screening of EOs and extracts by chromatographic techniques (GC-MS, HPLC, LC-MS). There are also coumarin and amino acids. Depending on the origin of the plants, the concentration and structure of the predominant chemicals vary significantly from one sample to another, establishing different chemotypes.

The pharmacological investigation of *M. chamomilla* was attributed to the chemical composition, containing numerous biocompound types.

The most important application of *M. chamomilla* was in the medicinal field on animal models and on human patients; the results showed the therapeutic effect of this plant on a wide range of diseases, including nervous cardiovascular, gastrointestinal, skin and reproductive diseases, obesity and related metabolic disorders, allergies, eye dysfunctions, acting as a protective agent in kidney, liver, among other systems.

## Figures and Tables

**Figure 1 life-12-00479-f001:**
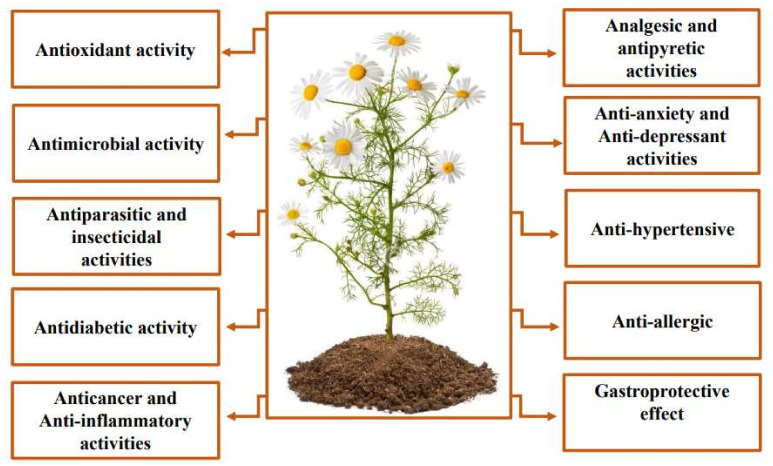
Biological properties of *Matricaria chamomilla*.

**Figure 2 life-12-00479-f002:**
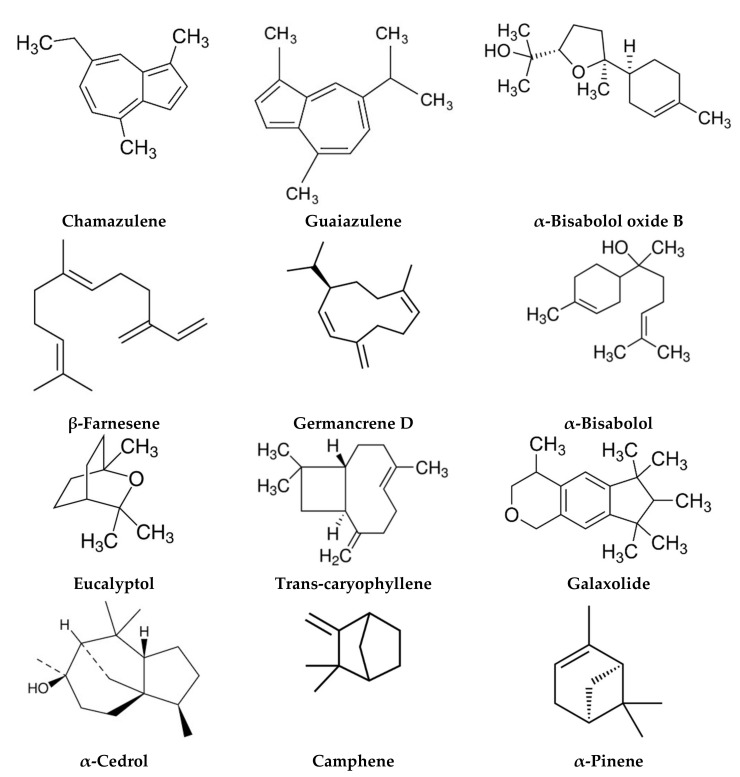
Structure of terpenoids identified in *Matricaria chamomilla*.

**Figure 3 life-12-00479-f003:**
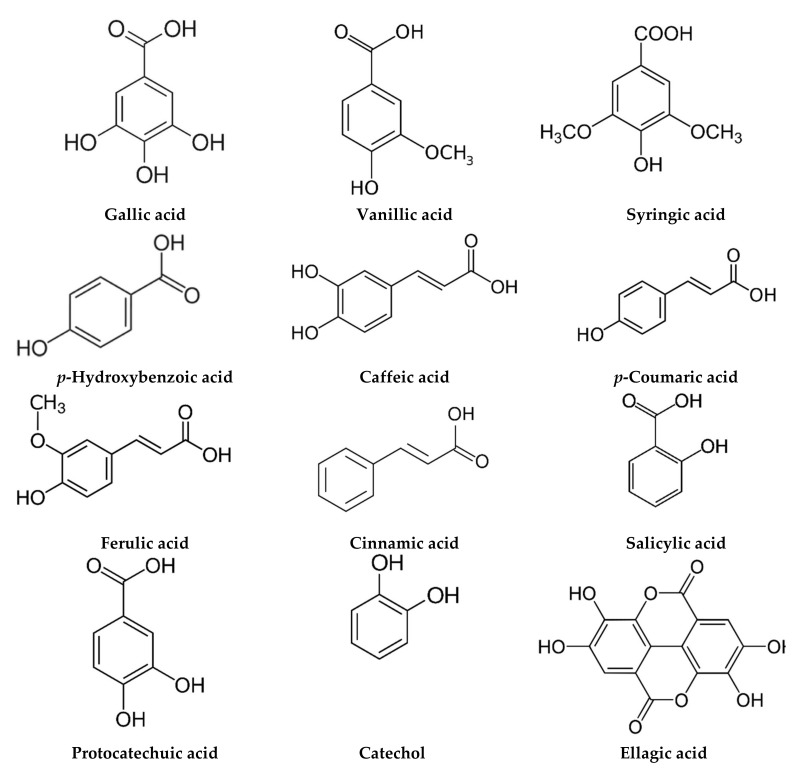
Structure of phenolic compounds identified in *Matricaria chamomilla*.

**Figure 4 life-12-00479-f004:**
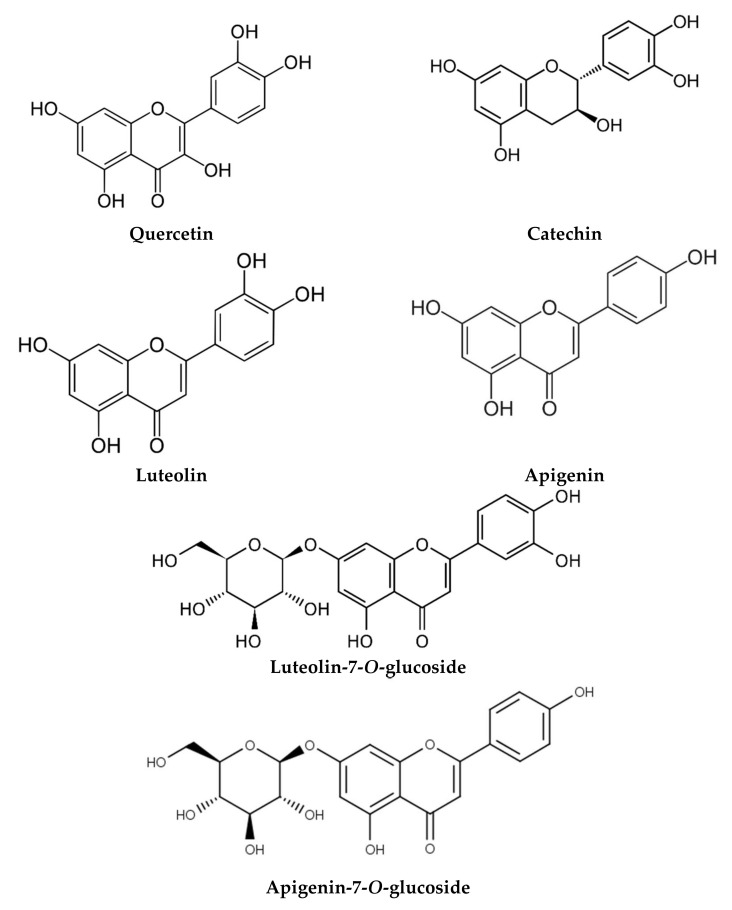
Structure of flavonoids identified in *Matricaria chamomilla*.

**Figure 5 life-12-00479-f005:**
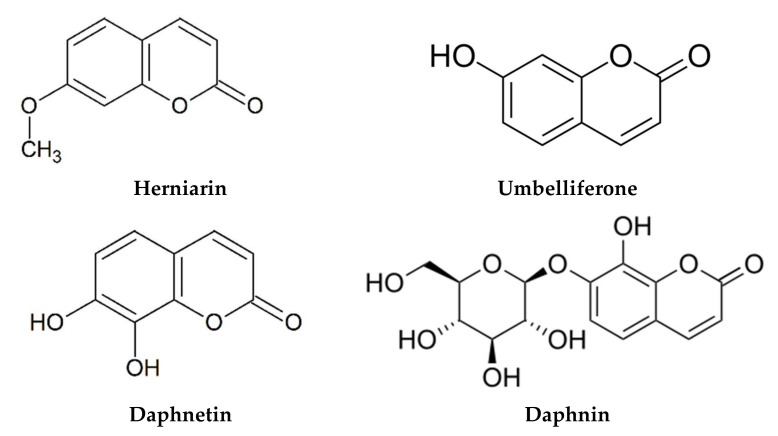
Structure of coumarins compounds identified in *Matricaria chamomilla*.

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
