# Peer review of "Chamomile (Matricaria chamomilla L.): A Review of Ethnomedicinal Use, Phytochemistry and Pharmacological Uses"

_life, 2022, doi:10.3390/life12040479_

Round 1

Reviewer 1 Report

Dear Authors 

After reviewing your "review paper" I can say that it is well written, exclusive and informative. English is very good as  well as referencing (number or reference and style). 

I have few comments on the paper:

  1. The topic is not "hot" and many studies were conducted on Chamomile
  2. A similar review paper was published very recently. You can check it on the following: Plants 202211(1),29; https://doi.org/10.3390/plants11010029.
  3. In page 9, two compounds have the same name (ß-farnesene). Please check and correct. 

Author Response

After reviewing your "review paper" I can say that it is well written, exclusive and informative. English is very good as well as referencing (number or reference and style). 

Thank you very much for your comments and suggestions.

I have few comments on the paper:

  1. The topic is not "hot" and many studies were conducted on Chamomile

R: This is a very interesting topic always. Our review apport new and interesting traditional uses, botanical and pharmacological studies in vitro and in vivo.

  1. A similar review paper was published very recently. You can check it on the following: Plants202211(1),29; https://doi.org/10.3390/plants11010029

R: The review of Chauhan et al. is an excellent work but differ in many aspects of our review. For example, Chauhan et al. have been concentrated in agrotechnological aspects. Also, in safety and legal aspects. However, our work emphasizer in medical and pharmacological applications in vivo and in vitro.

  1. In page 9, two compounds have the same name (ß-farnesene). Please check and correct.

R: Thank you very much for you remarque, we have checked the figure and corrected it.

Reviewer 2 Report

Dear authors,

The article is very well written and provides significant data regarding the ethnomedicinal use, phytochemistry and therapeutic properties of M. chamomilla. Below you can find the modifications to the manuscript.

Line 22: Google Scholar were used to gather data 

Line 23: showing that the plant contains over 120

Line 42: It is an annual herb that grows on all soil types and is resistant to cold

Line 143: In Turkey

Line 187: high variation in component quantity in EOs

Line 219: and phenolic acids (benzoic and rosmarinic acids)

Line 671: Traditionally, M. chamomilla was used to

Author Response

The article is very well written and provides significant data regarding the ethnomedicinal use, phytochemistry and therapeutic properties of M. chamomilla. Below you can find the modifications to the manuscript.

I agree with your comments.

Line 22: Google Scholar were used to gather data 

Line 23: showing that the plant contains over 120

Line 42: It is an annual herb that grows on all soil types and is resistant to cold

Line 143: In Turkey

Line 187: high variation in component quantity in EOs

Line 219: and phenolic acids (benzoic and rosmarinic acids)

Line 671: Traditionally, M. chamomilla was used to

R: Thank you for your typo’s corrections. All have been incorporated.

Reviewer 3 Report

This article is not suitable to review due to its lack of novelty.

Author Response

This article is not suitable to review due to its lack of novelty.

R: See the document attached.

Reviewer 4 Report

The 1633754 manuscript is a review about ethnomedicinal, phytochemistry and pharmacological uses of chamomile. This review included an exhaustive number of articles (218) related to this plant. The manuscript was well organized including i) the botanical and ethnomedicinal uses of chamomile, ii) the phytochemical analyzes of extracts and essential oils obtained from this vegetal species, iii) the in vivo and in vitro pharmacological studies carried out with extracts and essential oils of chamomile and iv) the encapsulation of essential oils and extracts was also exposed. In addition, the information obtained from the bibliography was well analyzed. Although the information in the articles was widely exposed, the relationship between the components of essential oils (extracts) and their biological activities could have been mentioned. Finally, conclusions were adequately presented. 

Author Response

R: Thank you very much for your positive comments. In respect to the relationship between chemical components and biological activities, currently, it is very difficult to establish a direct relation between a specific component and some biological activity, because generally EOs and herb extracts are very complexes, being formed by tens of possible active substances.

Round 2

Reviewer 3 Report

Accept in this current form.